# MACEval: A Multi-Agent Continual Evaluation Network for Large Models

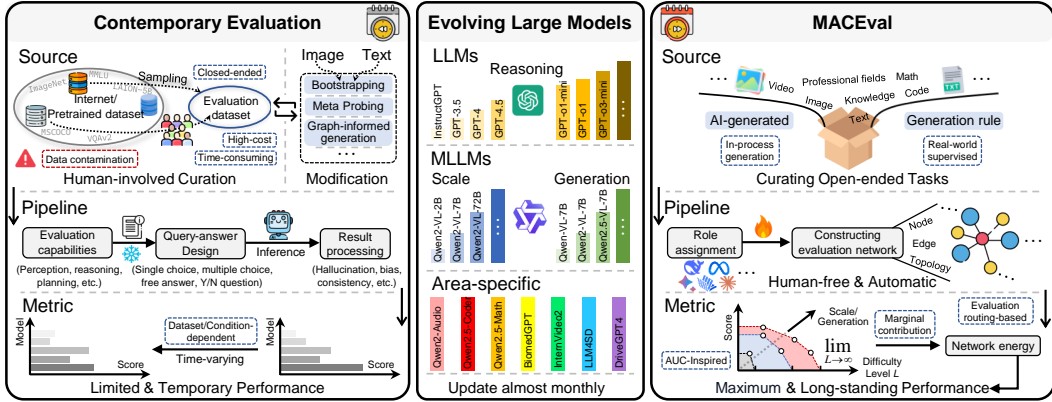

Figure 1: Paradigm comparison between current large model evaluations and our proposed **MACEval**.

## ABSTRACT

Hundreds of benchmarks dedicated to evaluating large models from multiple perspectives have been presented over the past few years. Albeit substantial efforts, most of them remain closed-ended and are prone to overfitting due to the potential data contamination in the ever-growing training corpus of large models, thereby undermining the credibility of the evaluation. Moreover, the increasing scale and scope of current benchmarks with transient metrics, as well as the heavily human-dependent curation procedure, pose significant challenges for timely maintenance and adaptation to gauge the advancing capabilities of large models. In this paper, we introduce **MACEval**, a **M**ulti-**A**gent **C**ontinual **Eval**uation network for dynamic evaluation of large models, and define a new set of metrics to quantify performance longitudinally and sustainably. MACEval adopts an interactive and autonomous evaluation mode that employs role assignment, in-process data generation, and evaluation routing through a cascaded agent network. Extensive experiments on 9 open-ended tasks with 23 participating large models demonstrate that MACEval is (1) human-free and automatic, mitigating laborious result processing with inter-agent judgment guided; (2) efficient and economical, reducing a considerable amount of data and overhead to obtain similar results compared to related benchmarks; and (3) flexible and scalable, migrating or integrating existing benchmarks via customized evaluation topologies. We hope that MACEval can broaden future directions of large model evaluation.

## 1 INTRODUCTION

Large Language Models (LLMs) and Multimodal Large Language Models (MLLMs) have achieved unprecedented performance in diverse applications such as visual understanding (Achiam et al., 2023; Kavukcuoglu, 2025; Zhu et al., 2025), complex long-text reasoning (math and coding) (Anthropic, 2024; Grattafiori et al., 2024; Team, 2025), and spatial cognition (Wu et al., 2024b; Yang et al., 2024b). Evaluation holds an equally pivotal role as pre-training in the development of large models, functioning not only as a benchmark for understanding current performance but also as a critical

pillar for aligning model capabilities with intended objectives. In this regard, it is crucial to construct transparent, trustworthy, and zero-contamination evaluations.

Despite the proliferation of various evaluation benchmarks, doubts about the genuine capabilities of large models persist (Li et al., 2024d; Yang et al., 2024c), particularly when discrepancies arise between leaderboard scores and real-world user experience. As shown in Fig. 1, we conclude the following three fundamental issues from the perspectives of data source, evaluation pipeline, and evaluation metric: **1) Closed-ended data.** Most benchmarks, alongside a corresponding well-organized dataset, which contains evaluation materials (*e.g.*, texts, images, or other modalities) collected from the Internet or existing large-scale datasets, pose potential data contamination risks of overlap with the training set (Xu et al., 2024; Deng et al., 2024). Once constructed, such static datasets are susceptible to obsolescence due to overfitting, especially considering the current rapid monthly update cycle of large models; **2) Human-dependent evaluation pipeline.** As the scale and coverage of current benchmarks continue to expand, the drawbacks of heavily human-involved annotation (*e.g.*, query-answer design and data refinement) and result review that traditional benchmarks (Wu et al., 2024a; Liu et al., 2024b; Yue et al., 2024a) rely on are gradually being exposed. Its time-consuming and high-cost characteristics hinder timely maintenance and updates to catch up with the models' evolving performance; **3) Limited and temporary metric.** The current performance quantification strategies that are limited by closed-ended datasets may not reflect the model's maximum capabilities. Besides, many metrics are transient in nature and fail to incrementally reflect performance changes for future models, making longitudinal assessment challenging. Recently, some researchers have proposed to address data-related issues by employing techniques such as vision-language bootstrapping (Yang et al., 2024c), establishing specific data generation rules (Zhu et al., 2024a), or meta probing agent (Zhu et al., 2024b). However, solutions for improving evaluation pipelines and metrics remain largely unexplored.

To address these problems, we introduce **MACEval**, a dynamic continual evaluation framework that measures the progress of large models autonomously by implementing a multi-agent collaboration system. First, we model the traditional one-way problem-solving evaluation process as an interactive interview process and assign three distinct roles in MACEval, *i.e.*, the interviewee, the interviewer, and the supervisor, who are responsible for answering, developing the questions, and inspecting the whole evaluation process, respectively. In particular, adhering to the design principles of open-ended evaluation tasks, we deploy reliable and contamination-free data sources by integrating an in-process generation strategy with real-world rule-based supervision, thus reducing unnecessary data collection and evaluation expenditures. Furthermore, instead of relying on human procedures, MACEval builds an agent-based evaluation network with a message-passing mechanism to automatically perform assessments across different tasks through pre-defined evaluation topologies, which enables hierarchical capability evaluation and is more flexible. Based on this, we propose an Area Under Curve (AUC)-inspired evaluation routing-based metric to measure the overall performance of a large model continually and maximally. Experiments on 23 popular large models across five typical capabilities demonstrate the effectiveness and efficiency of the proposed MACEval.

Our contributions are summarized in three-fold:

- We introduce MACEval (Fig. 2), a multi-agent continual evaluation network for dynamic large model assessment. By modeling the evaluation procedure as an autonomous interview, MACEval effectively mitigates data contamination and enhances evaluation efficiency.

- We propose an AUC-inspired quantitative performance metric for large models to evaluate longitudinally, which can be further incorporated with the evaluation routing to assess the overall performance across various capabilities.

- We evaluate comprehensively the proprietary and open-source LLMs and MLLMs on several open-ended tasks to demonstrate the effectiveness of the proposed MACEval. Evaluation results show superior efficiency, flexibility, and scalability of MACEval compared to existing benchmarks, thus offering valuable directions for future research on large model evaluation.

## 2 RELATED WORK

**Benchmarks for Large Models.** Benchmarks are the foundations of applied large generative model research. Over the past few years, significant efforts have been made to evaluate LLMs and MLLMs

from multiple perspectives (Chang et al., 2024; Li et al., 2024c). From textual comprehension (Bai et al., 2023; Bang et al., 2023) and visual perception (Liu et al., 2023; Fu et al., 2024; Li et al., 2024a) to multistep reasoning (Shao et al., 2024; Chen et al., 2024b; Rajabi & Kosecka, 2024) and other specialized domains (Bai et al., 2024; Chen et al., 2025; Yue et al., 2024a; He et al., 2024), evaluation results not only serve as practical guidance for end users but also provide developers with actionable insights for model optimization. With the rapid development of large models and their real-time Internet data crawling strategy used for training, most benchmarks face the risk of being quickly overfitted. Besides, the static, labor-intensive nature and sheer scale of existing benchmarks aggravate the difficulty of maintenance and timely updates (Banerjee et al., 2024), thereby enhancing their susceptibility to data contamination. Therefore, there is an urgent need to establish a long-lasting, dynamic, and robust evaluation framework for large models.

**Data Contamination.** Recently, data contamination issues have gained widespread attention across evaluation benchmarks for both LLMs and MLLMs (Carlini et al., 2023; Xu et al., 2024; Li & Flanigan, 2024). Researchers from OpenAI and the Llama group conducted contamination studies for GPT-4 (Achiam et al., 2023) and Llama2 (Touvron et al., 2023) on their pre-training data. Study (Li et al., 2024d) reveals significant contamination rates in academic exam-based benchmarks such as MMLU (29.1%) (Hendrycks et al., 2021) and C-Eval (45.8%) (Huang et al., 2023), primarily attributed to the widespread dissemination and circulation of academic test questions. Deng *et al.* (Deng et al., 2024) proposed a corpora overlap investigation protocol, TS-Guessing, and detected 57% exact match rate of ChatGPT in predicting masked choices in the MMLU test set. Yang *et al.* (Yang et al., 2024c) reported image overlaps of over 84.4% and 33.2% between SEEDBench (Li et al., 2024a) and pre-training datasets LAION-100M (Schuhmann et al., 2021) and CC3M (Sharma et al., 2018), respectively. Such exposure can significantly impact the reliability of the evaluation, leading to inflated results that do not accurately reflect the true capabilities of the large models. In this paper, we address this problem by reforming the source of evaluation content, introducing AI-generated data and an open-ended task design strategy to enhance benchmark robustness.

**Dynamic Evaluation.** One recent promising attempt to mitigate the issue of data contamination in large model evaluations is dynamic evaluation. Wang et al. (2025) implemented several perturbations, such as paraphrasing, adding noise, and reversing polarity, to construct evolving instances for testing LLMs against diverse queries and data noise. DyVal (Zhu et al., 2024a) proposes to dynamically generate evaluation samples under pre-defined constraints, which modulates a graph-based algorithm generation structure and fine-grained control over problem difficulty by adjusting the structural complexity. Afterwards, Zhu et al. (2024b) presented meta-probing agents, which automatically refresh an original evaluation problem following psychometric theory on three basic cognitive abilities, including language understanding, problem solving, and domain knowledge. In (Yang et al., 2024c), the authors designed various bootstrapping strategies (*e.g.* image editing and sentence rephrasing) with complexity control for both image and question modification. However, these studies primarily focus on modifying the sources of evaluation data, without achieving fully dynamic and automatic evaluation processes. Moreover, the corresponding evaluation metrics remain static and transient, failing to adequately reflect the dynamic evolution characteristics of large models.

## 3 THE MACEVAL

### 3.1 DESIGN PRINCIPLES

**Focusing on Autonomous Evaluation Process.** Unlike existing LLM or MLLM benchmarks that are strongly human-involving in source content collection and relatively independent in cross-ability evaluation, our MACEval follows two basic principles: **(1)** Not requiring pre-collected evaluation datasets, and all visual or text query-answer pairs are dynamically generated during the process; **(2)** Progressive capability evaluation scheme with real-time task adjustment. We adhere to the principles in building the MACEval, while making it applicable to evaluating all-round abilities.

**Pushing to the Limit.** Since existing benchmark datasets are finite and closed-ended, the measured performance scores do not reflect the model's maximum capabilities. To push the evaluated model to the limit, we adopt a stress-testing strategy in which the model is continuously challenged with increasingly difficult query-answer tasks until it fails to provide a correct response. Henceforth, we can derive a long-standing performance metric by iteratively updating the envelope area formed by connecting performance points across different difficulty levels.

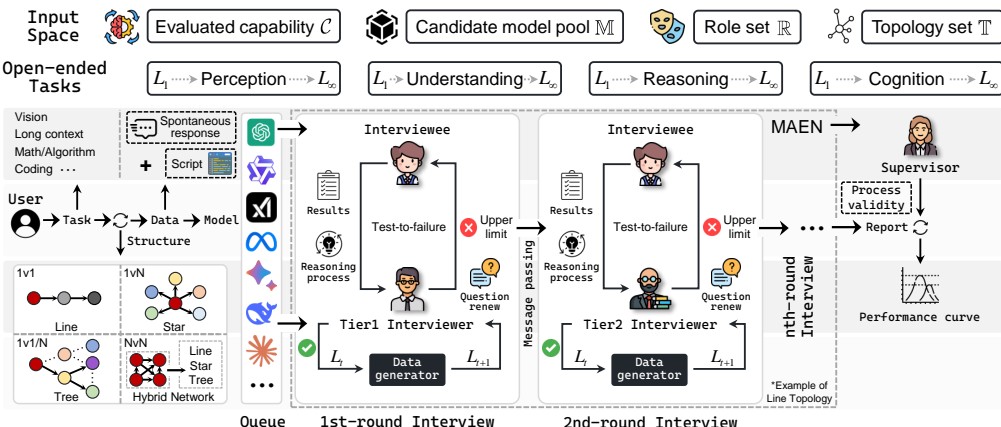

Figure 2: An overview of our proposed MACEval, which consists of three primary phases: evaluation capability determination, MAEN construction, and open-ended task selection. The pipeline models the evaluation of large models as a multi-round interview process. Specialized agents like interviewers for direct performance evaluation and third-party supervisors for validity assessment of the entire process with a message propagation mechanism, enabling collaboration between interviewer models and other functional models to efficiently and automatically produce reliable evaluation outputs.

## 3.2 CURATING OPEN-ENDED TASKS

Previous benchmarks for large models primarily focus on collecting raw materials from the Internet or existing general datasets to construct evaluation datasets, leading to potential data contamination risks due to overlaps between evaluation sets and the training data of large models (Touvron et al., 2023; Zhu et al., 2024a; Yang et al., 2024c). In this paper, we avoid such a problem by curating open-ended tasks with quantitatively adjustable difficulty for evaluation and adopt an in-process generation manner through the multi-agent evaluation strategy. This ensures the randomness of the evaluation process while guaranteeing the uniqueness of the target results under the given task. Specifically, we conduct a preliminary exploration of 9 tasks across five key domains currently emphasized in the field: visual perception, textual comprehension, math, algorithms, and coding (White et al., 2025), as listed in Tab. 1. See Appendix B for additional details.

**Visual Perception.** Evaluating the visual perception capabilities of MLLMs has always been one of the foundational aspects of recent research in MLLMs, featuring prominently in many releases and serving as a key reference for downstream task applications (Bai et al., 2025; Zhu et al., 2025). As illustrated in Fig. 3, we evaluate the low-level and high-level perception capabilities (Wu et al., 2024a) via an Image Quality Perception (IQP) task and a Content Understanding (CU) task, respectively. For the IQP task, we dynamically generate a set of Gaussian white noise with different intensities by increasing the variance and adding it to the original images. For the CU task, we query the MLLM to identify the number of repeated icons in images with progressively increasing grid sizes.

**Text Comprehension.** For the most fundamental text comprehension task in LLM evaluation, we include a Scrambled Text Understanding (STU) task, *i.e.*, inferring the meaning of a sentence while ignoring typos and misspellings, and a String Parsing (SP) task that evaluates the fine-grained character-level textual processing abilities. Concretely, for the STU task, we disrupt or mask the word structure to generate scrambled text. The proportion of perturbed words relative to the total number of words in the sentence serves as a variable for controlling the difficulty level. For the SP task, the length of the string, namely the number of irrelevant characters, serves as the difficulty variable.

**Math.** Mathematical reasoning is a cornerstone for assessing the ability of LLMs to resolve complex problems and make multi-hop reasoning, which plays a significant role in general LLM research (Yang et al., 2023; Yue et al., 2024b). Current math-specific benchmarks (Zhang et al., 2024; Xia et al., 2025) mainly collect diverse sets of problems from major textbooks and online resources, such as geometry, linear algebra, and calculus, but rarely provide a quantitative grading of problem difficulty, thereby hindering their direct applicability in the design of open-ended evaluation tasks. Here, we consider the basic arithmetic calculations and encompass two intuitive difficulty control means, *i.e.*, changing the numerical scale and the number of operations, as shown in Fig. 3.

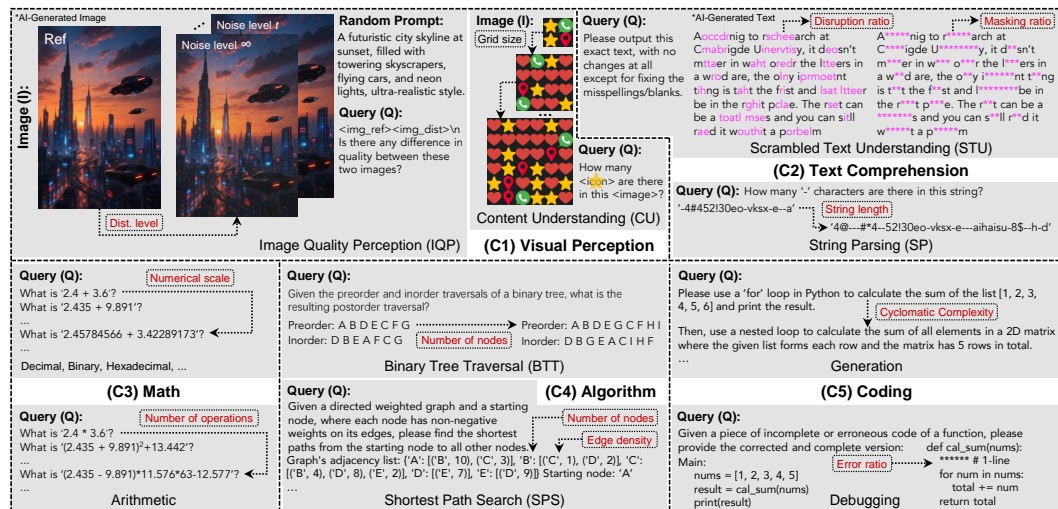

Figure 3: A data card of 9 open-ended tasks that evaluate the visual perception, text comprehension, math, algorithm reasoning, and coding abilities of large models.

**Algorithm.** As a derivative task of basic mathematics, solving algorithmic problems constitutes another important aspect for assessing the reasoning capabilities of LLMs. Specifically, we select two common tasks in computer science, Binary Tree Traversal (BTT) and Shortest Path Search (SPS), for evaluation. Among them, the number of nodes and the edge density that determine the complexity of the data structure are taken as the variables.

**Coding.** Coding is among the most practical capabilities of LLMs in real-world scenarios. Following (Chen et al., 2024c; Jain et al., 2025), we design two open-ended code generation and debugging tasks with dynamic difficulty. The generation task specifically focuses on the ability of models to parse a cyclomatic complexity evolved coding question statement and write an executable answer. The debugging scenario assesses the self-repair capabilities of LLMs. Here, the model is given a problem statement and its corresponding code with partially removed content (*missing ratio*), and then prompted to generate a repaired version.

Table 1: Summary of 9 open-ended tasks across 5 capabilities. (IQP: Image Quality Perception; CU: Content Understanding; STU: Scrambled Text Understanding; SP = String Parsing; BTT: Binary Tree Traversal; SPS: Shortest Path Search.)

| Capability | Task | Variable | Target |
|---|---|---|---|
| Visual Perception | IQP (*low-level*) | Distortion level | MLLM |
| | CU (*high-level*) | Grid size | MLLM |
| Text Comprehension | STU | Disruption ratio | LLM |
| | | Masking ratio | LLM |
| | SP | String length | LLM |
| Math | Arithmetic | Numerical scale | LLM |
| | | Number of operations | LLM |
| Algorithm | BTT | Number of nodes (layer) | LLM |
| | SPS | Number of nodes | LLM |
| | | Edge density | LLM |
| Coding | Generation | Cyclomatic complexity | LLM |
| | Debugging | Missing ratio | LLM |

### 3.3 DEFINITION AND FORMULATION

We define the input space of a multi-agent system as $\mathbb{I} = (\mathbb{M}, \mathbb{R}, \mathbb{T})$, where $\mathbb{M}$ denotes the candidate pool of LLMs or MLLMs. $\mathbb{R}$ represents the set of pre-defined agent roles (*e.g.*, interviewee, interviewer, and supervisor), and $\mathbb{T}$ is the set of evaluation topologies, *i.e.* collaboration modes, such as point-to-point link, tree, star, and multi-hop network. Within the evaluation space, a Multi-Agent Evaluation Network (MAEN) instance is defined as follows:

**Definition 1 (Multi-Agent Evaluation Network)** *The graph-based MAEN $\mathcal{G}$ includes several large models with distinct identities (node type), participating collaboratively (edge type) in a configurable (topological type) evaluation process:*

$$\mathcal{G} = \left\{ \{\mathcal{M}_i\}_{i=1}^{M}, \{\mathcal{R}_j\}_{j=1}^{R}, \{\mathcal{T}_k\}_{k=1}^{T} \right\}, \quad \mathcal{M}_i \in \mathbb{M}, \ \mathcal{R}_j \in \mathbb{R}, \ \mathcal{T}_k \in \mathbb{T}, \quad (1)$$

*where $\mathcal{M}$, $\mathcal{R}$, and $\mathcal{T}$ correspond to the selected large model, role function, and evaluation network structure, respectively. $M$, $R$, and $T$ are the number of each corresponding component.*

**Definition 2 (Multi-Agent Continual Evaluation Stream)** *The MACEval stream can be denoted by a mapping function $f : \mathbb{M} \times \mathbb{R} \times \mathbb{T} \to \mathcal{G}$ that maps the input space $\mathbb{I}$ to an MAEN $\mathcal{G}$ tailored for the evaluated capability $\mathcal{C}$ from a non-stationary data stream $\mathcal{S} = \{(q, a)_0, (q, a)_1, \cdots, (q, a)_\infty\}$, where data tuple $(q, a)_t$ (query-answer pair) is dynamically generated via a step $t$-dependent function.*

**Definition 3 (Message Passing)** *Given a MAEN $\mathcal{G}$, an interviewee node $\mathcal{M}^E$, a set of downstream adjacent interviewer nodes $\mathcal{U} = \{\mathcal{M}_1^I, \cdots, \mathcal{M}_N^I\}$, and the message (dialogues during the evaluation process) $\mathcal{D} = \{d_1^t, \cdots, d_N^t\}$, where $N$ is the size of $\mathcal{U}$, and $d_1^t$ denotes the message generated during the interaction between $\mathcal{M}^E$ and $\mathcal{M}_1^I$ at step-$t$, message passing aggregates all the messages $\mathcal{D}$ to higher-tier interviewer nodes $\tilde{\mathcal{M}}^I$ to produce updated data, which can be formulated as:*

$$(\hat{q}, \hat{a})_{t+1} = \tilde{\mathcal{M}}^I\left((q, a)_t, \mathcal{D}\right). \tag{2}$$

*Specifically, when the message passing terminates at a supervisor node, the aggregated messages store the information of the entire MAEN, which reflects the validity of the evaluation process.*

### 3.4 A FRAMEWORK FOR MACEVAL

The proposed MACEval network is configured by three main factors: **1)** the role assignment defines the basic functionality of network nodes; **2)** the evaluation topology determines the sequential order and interdependencies within the evaluation process; **3)** the evaluation network-based metrics and the evaluating continually mechanism provides comprehensive and sustainable capability assessment.

**Role Assignment.** As shown in Fig. 2, the evaluation process for large models is conceptualized as an interview-like procedure, where tasks are posed sequentially or in parallel for probing different capabilities. We mainly involve three types of agents, *i.e.*, the interviewer, the interviewee, and the supervisor. Among them, the interviewee, namely the evaluated model, generates responses according to the given queries. The interviewer performs real-time performance evaluations and determines whether to transform a given question into a new one with dynamically increasing difficulty based on the interviewee's responses. To ensure the validity of the evaluation process and to avoid scenarios such as repetitive questioning or posing questions that do not meet evaluation requirements, we deploy a third-party model distinct from the evaluation and interviewer models to serve as a supervisory component.

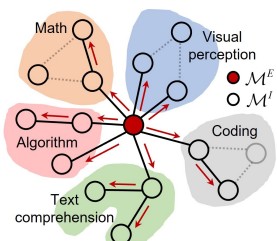

Figure 4: Example of an evaluation network, where colored clusters represent evaluation agents for different capabilities and red arrows denote activated evaluation routes.

**Evaluation Topology.** Prior benchmarks run each task independently and average the task metrics on separate subsets of the collected dataset to get an overall metric, neglecting the inherent dependencies between tasks while failing to support progressive and sustainable evaluation (Chang et al., 2024). In MACEval, we introduce four evaluation topologies, including line, star, tree, and hybrid network, to establish the relationships among different evaluation streams. This also enables users to construct customized evaluation networks that better align with the assessment requirements of their target capability. More details are in Appendix C.

**Evaluating Continually.** During the evaluation phase, an interviewer continually updates the data tuples $(q, a)_t$ based on new responses from the interviewee model, until it fails to answer correctly at a given level of difficulty. After that, the interviewee model, together with its performance records from the previous round, enters the next interview round. Such a spontaneous evaluation process goes beyond the limitations imposed by fixed dataset size, enabling deeper exploration of the model's extreme performance while avoiding the manual collection of excessive and redundant data. Given the above objectives, we design an Area Under Curve (AUC)-inspired metric:

$$\text{ACC-AUC} = \int_a^{\arg\min_t \{t | \text{ACC}(t)=0\}} \text{ACC}(t)dt, \quad \text{ACC}(t) = \frac{\sum_{r=1}^{\mathcal{Q}_{total}} \mathbf{1}\left\{\mathcal{M}^E\left(q_{r,t}\right) \cong a_{r,t}\right\}}{\mathcal{Q}_{total}}, \tag{3}$$

where $a$ is the initial difficulty level, and $\mathbf{1}\{\cdot\}$ represents the indicator function, which returns 1 if the condition inside the braces holds, and 0 otherwise. $q_{r,t}$ and $a_{r,t}$ are the $r$-th query-answer pair at difficulty level $t$. $\mathcal{Q}_{total}$ denotes the total number of query-answer pairs at a certain level. This simple and generally-applicable strategy enables us to evaluate the large models *longitudinally* and *sustainably*, as verified in our experimental analysis (Fig. 5).

### 3.5 EVALUATION ROUTING-BASED METRIC

Building upon the single-dimensional evaluation strategy discussed previously, we introduce a novel evaluation routing-based metric to measure the overall capability of large models. As illustrated in

Table 2: The performance of different interviewees (LLMs) on 4 text-only capabilities under a 1-hop line evaluation topology configuration. We report ACC-AUC and the difficulty level corresponding to the maximum performance, separated by a slash. The best results are highlighted in bold.

| Interviewee (LLM) | Interviewer | Text Comprehension | | | Math | | Algorithm | | | Coding | |
|---|---|---|---|---|---|---|---|---|---|---|---|
| | | $STU_{disrupt.}$ | $STU_{mask.}$ | SP | $Arith._{scale}$ | $Arith._{oper.}$ | BTT | $SPS_{node}$ | $SPS_{edge}$ | Gener. | Debug. |
| GPT-4o | GPT-4o | 8.857/10 | 6.043/**7** | 2.6/6 | 1.7/2 | 0.4/1 | 3.1/4 | 5.5/10 | 4.1/6 | 6.1/11 | 14.3/16 |
| GPT-4.1 | | 8.438/9 | 4.815/6 | 3.6/**10** | 1.8/2 | 0.3/1 | 2.6/4 | 5.2/10 | 3.7/6 | 6.3/12 | 14.8/16 |
| Gemini 1.5 Pro | GPT-4o | 8.766/10 | 5.289/6 | 1.4/3 | 1.3/2 | 0.5/1 | 2.8/3 | 5.4/10 | 4.2/6 | 6.1/12 | 14.6/18 |
| Gemini 2.0 Flash | | 8.898/10 | 5.049/6 | 2.8/8 | 1.6/2 | 0.6/1 | 3.1/4 | 6.1/10 | 4.5/7 | 7.3/13 | 16.6/22 |
| Gemini 2.5 Pro | | 8.978/10 | **6.062/7** | 3.8/9 | 3.3/5 | **0.9/3** | **3.2**/4 | 6.2/10 | 4.9/8 | **7.8/15** | 17.8/22 |
| DeepSeek-V3 | GPT-4o | 5.293/6 | 5.099/6 | 2.0/5 | 2.6/4 | 0.7/1 | 3.0/3 | 4.4/10 | 4.6/6 | 6.4/13 | 15.3/17 |
| DeepSeek-R1 | | 9.871/10 | 5.936/**7** | **8.3/10** | **7.2/10** | **0.9**/1 | 3.1/**5** | **9.3/13** | **7.2/9** | 7.5/13 | **17.9**/20 |
| Qwen3-8B | GPT-4o | 7.626/10 | 2.846/4 | 1.4/4 | 1.0/1 | **0.9/2** | 1.1/2 | 4.2/8 | 2.8/4 | 5.8/8 | 11.5/14 |
| Qwen2.5-7B | | 6.797/8 | 3.003/4 | 0.9/2 | 0.9/1 | 0.5/1 | 1.6/2 | 3.4/8 | 2.6/4 | 5.2/8 | 9.8/13 |
| Qwen2.5-14B | | 7.903/10 | 3.327/4 | 2.0/5 | 1.0/1 | 0.5/1 | 2.2/3 | 4.4/8 | 3.0/4 | 5.5/8 | 11.2/13 |
| Qwen2.5-72B | | 8.543/10 | 4.274/5 | 2.2/5 | 1.0/1 | 0.5/1 | 2.4/3 | 2.8/4 | 3.7/5 | 5.8/9 | 13.9/15 |
| Qwen2-7B | | 7.339/9 | 3.053/4 | 0.5/1 | 0.7/1 | 0.4/1 | 1.2/2 | 2.8/7 | 2.4/4 | 4.7/7 | 8.5/13 |
| Llama3.3-70B | GPT-4o | 7.619/10 | 4.564/6 | 2.8/8 | 1.0/1 | 0.5/1 | 2.6/3 | 4.1/8 | 3.3/4 | 5.7/9 | 13.6/14 |
| Llama3.2-3B | | 5.641/8 | 2.369/3 | 0.1/1 | 0.3/1 | 0.3/1 | 0.6/1 | 1.4/4 | 1.7/3 | 2.2/5 | 7.3/8 |
| Llama3.1-8B | | 6.740/9 | 3.006/4 | 0.7/4 | 0.2/1 | 0.4/1 | 0/0 | 2.9/8 | 2.1/4 | 4.6/7 | 9.3/12 |

Fig. 4, given an evaluation network with activated evaluation routing $\mathcal{P}$, the comprehensive ability of the interviewee model $\mathcal{M}^E$ can be measured as follows:

$$\epsilon_{\text{overall}} = \sum_{(\mathcal{M}^E, \mathcal{M}^I) \in \mathcal{P}} \text{ACC-AUC}_{(\mathcal{M}^E, \mathcal{M}^I)}. \tag{4}$$

This metric quantifies the overall model performance as the evaluation network energy derived from edge (*evaluation route*) weights. It considers the intrinsic dependencies among different capability assessments while enhancing user-oriented customization, thereby enabling a self-organizing and scalable evaluation framework.

## 4 EXPERIMENTS

### 4.1 EXPERIMENTAL SETUP

**Benchmark Candidates.** Our experiments include 23 large models total, with a mix of cutting-edge proprietary models, open-source LLMs, and MLLMs. Specifically, for proprietary models, we include OpenAI models such as GPT-4o (OpenAI, 2024), and GPT-4.1 (2025-04-14) (Achiam et al., 2023), Google models such as Gemini-1.5-Pro, Gemini-2.0-Flash (Reid et al., 2024), and Gemini-2.5-Pro (Kavukcuoglu, 2025). For open-source LLMs, we include three mainstream language backbones widely used in many large models, *i.e.*, the DeepSeeks (DeepSeek-V3 (Liu et al., 2024a) and DeepSeek-R1 (Guo et al., 2025)), the Qwens (Qwen3-8B (Team, 2025), Qwen2.5-{7, 14, 72}B (Team, 2024), Qwen2-7B (Yang et al., 2024a)), and the Llamas (Llama3.3-70B, Llama3.2-3B, Llama3.1-8B) (Grattafiori et al., 2024). For open-source MLLMs, we include models such as Qwen2.5-vl-{3, 7, 72}B (Bai et al., 2025), Qwen2-vl-7B (Wang et al., 2024), InternVL3-8B (Zhu et al., 2025), InternVL2.5-{8, 38}B (Chen et al., 2024d), InternVL2-8B (Chen et al., 2024e).

**Implementation Details.** For all models, we perform evaluation using their respective templates under zero shot settings. The initial difficulty level $a$ and the total number of queries in each level $\mathcal{Q}_{total}$ are set to 1 and 10, respectively. The max_new_tokens is set to the maximum value supported by each model to ensure the completeness of the response. We employ SSH connections to establish communication between the local interviewer model and the remote interviewee model running on a server. All experiments are conducted using a maximum of 8 RTX4090 24GB GPUs.

### 4.2 MAIN RESULTS

In Tab. 2 and Fig. 5, we analyze the performance of 15 LLMs and 8 MLLMs under different settings on 9 open-ended tasks. Our evaluation brings several important findings, as follows:

**1)** *Identifying the gap in upper-bound performance via continual evaluation.* As shown in Tab. 2, although the performance of different models varies, models of the same scale or generation generally share a similar upper limit of performance. This phenomenon is particularly evident in arithmetic and algorithmic tasks that require strong reasoning abilities, reflecting the common performance bottleneck of current LLMs.

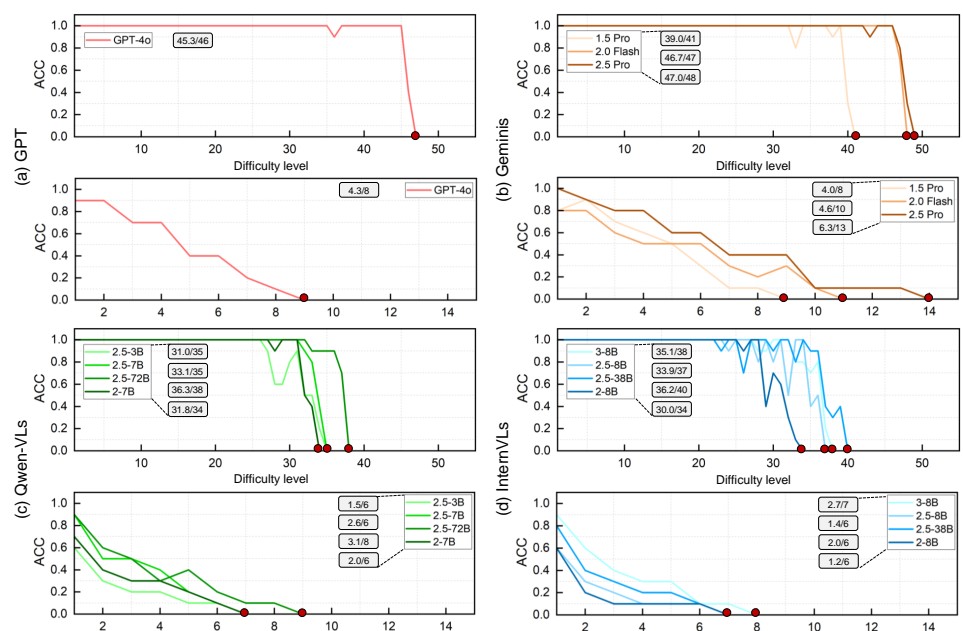

Figure 5: Performance curves and ACC-AUC values of different series of MLLMs. The positions of the red circles indicate the upper bound of the model's capabilities. The upper and lower parts of the subfigure depict the IQP and CU tasks, respectively.

**2)** *Models with downloadable weights currently lag behind the top-performing models.* All five proprietary models, especially the more advanced Gemini 2.5 Pro and GPT-4o, have shown superior performance to open-source models, such as the Qwen 2.5 and Llama 3.3 series, on most tasks. It is worth noting that DeepSeek-R1, with its powerful reasoning capabilities, outperforms proprietary models on over 60% of the tasks.

**3)** *Scaling law vs. Technological advancement.* Tab. 2 and Fig. 5 show a notable performance improvement in the evaluated capabilities when comparing larger models to smaller ones, as well as newer model generations to their predecessors, showing a relatively

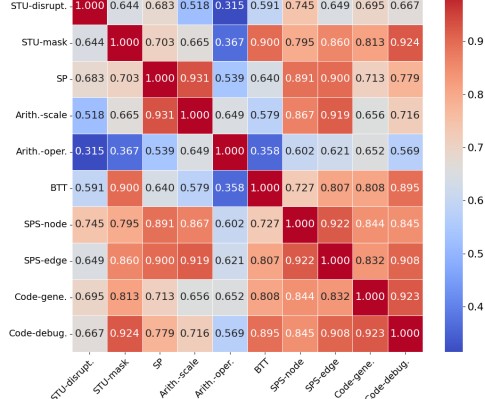

Figure 6: Correlations across different tasks.

strong scaling law. Meanwhile, the latest Qwen3-8B benefited from its thinking mode, significantly surpassing the previous Qwen2.5-14B, and almost on par with Qwen2.5-72B in math and coding.

**4)** *Steep decline in low-level visual perception.* In Fig. 5, we observe a sharp performance cliff in the low-level IQP tasks, which suggests potential Just Noticeable Difference (JND) points in the vision backbones of MLLMs. Besides, the InternVL series exhibits more unstable perceptual responses compared to the Qwens and other proprietary models.

### 4.3 CORRELATION ANALYSES

In Fig. 6, we compute the Pearson correlation coefficient among all pairs of tasks. Results show that, unsurprisingly, text comprehension, algorithm reasoning, and coding all correlate with one another (avg. $r > 0.81$). Moreover, math tasks involving intensive numerical computation correlate relatively weakly with all other categories, which, together with the extremely low maximum capability level reported in Tab. 2, indicates the uniqueness of this capability. We further investigate the relationship between maximum capability level and long-term ACC-AUC in Fig. 7. It can be observed that these two dimensions are highly correlated (avg. $r \approx 0.888$) in all tasks, and few models reach saturated performance within a limited capability level.

### 4.4 EFFECT OF DIFFERENT EVALUATION TOPOLOGIES

**1)** *Message passing improves the quality of question generation.* We focus on two types of question generation errors caused by hallucinations from the interviewer model during the evaluation process,

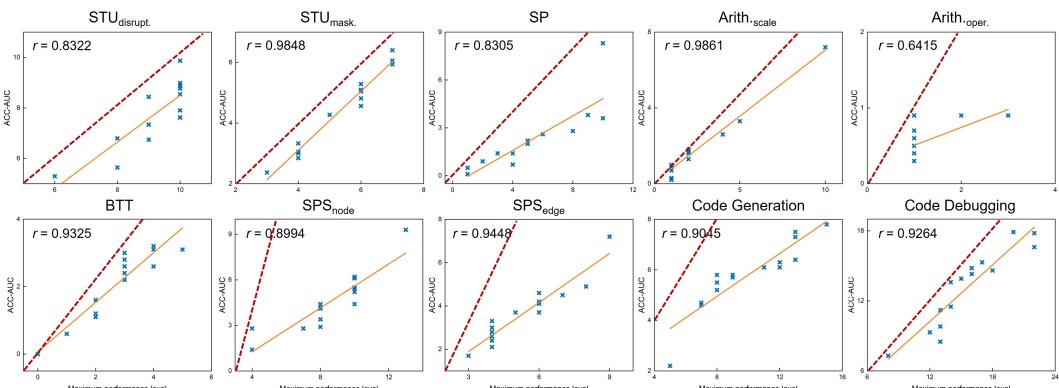

Figure 7: Maximum performance level vs. ACC-AUC. We plot the performance of 15 LLMs compared to the corresponding maximum difficulty levels with each a fit line and correlation coefficient. The red dotted lines denote the performance saturation line.

Table 3: Error rate of *w/* and *w/o* message passing (GPT-4o as interviewer). End nodes are in bold.

| Topology | $STU_{disrupt.}$ | $\textbf{STU}_{\textbf{mask.}}$ | SP | $Arith._{scale}$ | $\textbf{Arith.}_{\textbf{oper.}}$ | BTT | $SPS_{node}$ | $\textbf{SPS}_{\textbf{edge}}$ | Generation | **Debugging** |
|---|---|---|---|---|---|---|---|---|---|---|
| 1-hop line (*w/o*) | 36% | 42% | 18% | 2% | 50% | 52% | 26% | 36% | 30% | 18% |
| Node sequence | $STU_{disrupt.}$->$STU_{mask.}$ | | | $Arith._{scale}$->$Arith._{oper.}$ | | | $SPS_{node}$->$SPS_{edge}$ | | Generation->Debugging | |
| 2-hop line (*w/*) | 18% (-24%) | | | 4% (-46%) | | | 22% (-14%) | | 14% (-4%) | |

*i.e.*, format violations and redundant questions. Tab. 3 demonstrates the effectiveness of the message passing mechanism in improving the question quality between two related tasks with an average error rate reduction of 22%, which serves as a special prompt engineering. Since the subject of the visual perception tasks is the image itself, no obvious mistakes in question generation were detected.

**2)** *Overall performance comparison based on the evaluation network energy.* As shown in Fig. 10, we compute the overall performance for 15 LLMs based on the evaluation routes in Fig. 4. To ensure fairness, we remove two 1-hop visual perception-related evaluation nodes, which are not affected by the message-passing mechanism. We notice that proprietary models still maintain a dominant position. The characteristic of $\epsilon_{overall}$ lies in considering underlying inter-task dependencies, incorporating information-aware evaluation routing instead of independently assessing and aggregating task scores.

Table 4: Results of different interviewers.

| Interviewee | Interviewer | $STU_{mask.}$ | $Arith._{scale}$ | BTT | Debug. |
|---|---|---|---|---|---|
| Gemini 1.5 Pro | Gemini 2.5 Pro | 5.218/6 | 0.8/1 | 2.8/3 | 14.9/17 |
| | DeepSeek-V3 | 5.087/6 | 1.4/2 | 3.3/4 | 14.1/15 |
| | Qwen2.5-7B | 5.083/6 | 1.3/2 | 2.6/3 | 12.9/13 |
| Qwen3-8B | Gemini 2.5 Pro | 3.095/4 | 1.0/2 | 1.4/2 | 11.2/14 |
| | DeepSeek-V3 | 2.818/4 | 0.9/2 | 1.4/2 | 11.7/14 |
| | Qwen2.5-7B | 2.905/4 | 1.0/2 | 2.0/3 | 10.9/12 |
| Llama3.1-8B | Gemini 2.5 Pro | 2.946/4 | 0.1/1 | 0/0 | 9.6/12 |
| | DeepSeek-V3 | 2.930/4 | 0.3/1 | 0/0 | 9.4/12 |
| | Qwen2.5-7B | 2.998/4 | 0.2/1 | 0/0 | 9.0/11 |

### 4.5 ANALYSIS OF INTERVIEWER DIVERSITY

Next, we discuss the performance differences under different interviewer settings. As listed in Tab. 4, the evaluation results remain consistent across different interviewers in most scenarios. However, when evaluating tasks that involve more complex text generation, using a weaker model as the interviewer may limit the exploration of the interviewee model's upper-bound capabilities. See more experimental results in the Appendix. D.

## 5 CONCLUSION

This work introduces MACEval, a novel dynamic evaluation framework for large generative models, which provides effective solutions to avoid data contamination and reduce human participation by a multi-agent system-based automatic evaluation network. To identify the upper bound of performance and evaluate continually, we propose an AUC-inspired metric, further incorporating the evaluation topologies to measure the overall performance of large models. Evaluation results on five high-profile capabilities, including visual perception, text comprehension, math, algorithm reasoning, and coding, demonstrate the effectiveness, efficiency, and flexibility of MACEval compared to existing benchmarks. We believe that employing large models as agents to form the autonomous evaluation network presents a promising direction for achieving safe and fair evaluation.

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

APPENDIX

## A  LIMITATIONS AND SOCIETAL IMPACT

Although our study provides validation of the MACEval framework in various tasks, there are several potential limitations acknowledged below. First, our experiments are limited to 9 tasks across five common capabilities, encompassing a specific range of topics. Incorporating a broader spectrum of tasks, such as those within a specialty area, facilitates to yield a more comprehensive insights into MACEval's applicability. Second, we apply evaluation topologies with fewer hops, *i.e.*, nodes and edges, to preliminarily demonstrate the effectiveness of the proposed framework. In the future, it is possible to construct larger evaluation networks to include a richer array of capability assessments. Third, in MACEval, the evaluation data stream consists of question-answer pairs graded by difficulty. One of the principles in constructing open-ended tasks is that the difficulty should be quantifiable, which imposes limitations on the implementation of many tasks.

Meanwhile, our work holds the potential for significant societal impact, both positive and negative. First, the evaluation is normally considered as important as training in developing large models. An insightful evaluation can offer promising directions to large model developers. The main motivation of this work is to build automatic, scalable, and trustworthy evaluation framework for large models, aligning closely with the rapid iteration of current models. Our MACEval offer a promising direction that using large models to tackle the evaluation problems for themselves has a lower costs than the traditional human-dependent form on broad occasions. Second, the proposed interview-like autonomous evaluation pipeline with open-ended settings is more conducive to exploring the performance boundaries of large models with less subjective bias. Third, although our research is not directly related to the design of generative models, the possibility of spontaneously generating NSFW contents in other tasks due to hallucinations or ambiguity in the content of the dialogue cannot be ruled out, since we adopted an human-free autonomous multi-agent collaboration framework in the evaluation process, where the evaluation content is autonomously generated.

## B  DETAILED OPEN-ENDED TASKS CONSTRUCTION

The objective of curating open-ended tasks for large model evaluation is to mitigate data contamination issues. We design 9 tasks based on fully AI-generated and difficulty-controllable principles, spanning five capabilities, including visual perception, text comprehension, math, algorithm reasoning, and coding, to preliminarily validate the effectiveness of our MACEval framework:

- **Image Quality Perception (IQP):** First, we select a set of keywords (*people, animals, nature, plants, cars, objects, buildings, textures, sports, and interiors*) based on the categorization of the renowned image website `Unsplash`[1], which served as the source for generating image prompts. Then, we adopt an interviewer-assisted prompt generation strategy to obtain $\mathcal{Q}_{total}$ prompts. Five prevailing text-to-image models, including `Stable Diffusion-1.5` (Rombach et al., 2022), `Stable-Diffusion-3.5-medium` (AI, 2024), `Playground v2.5` (Li et al., 2024b), `FLUX.1-schnell` (Labs, 2024), and `Sana` (Xie et al., 2024), are randomly invoked externally to generate original image contents. The output image size is set to $512^2$ and the `num_inference_steps` is set to the default value. As for the difficulty control, we generate a set of Gaussian white noise by setting the variance to $[1 : \infty : 1]$ (which represents values starting from 1 with a step size of 1) and add it to the original images.

- **Content Understanding (CU):** This task evaluates the model's ability to distinguish image content. We download over 100 different icons from `FLATICON`[2] and randomly form part of them into a grid image, which contains smaller icons, as shown in Fig. 8. Then, we examine whether the interviewee model can correctly answer the exact number of target icons in the given grid images. The task difficulty is controlled by the number of grids, *i.e.*, the more the grids there are, the more interference terms are presented. We begin with the $3 \times 3$ size.

---

[1] https://unsplash.com/
[2] https://www.flaticon.com/

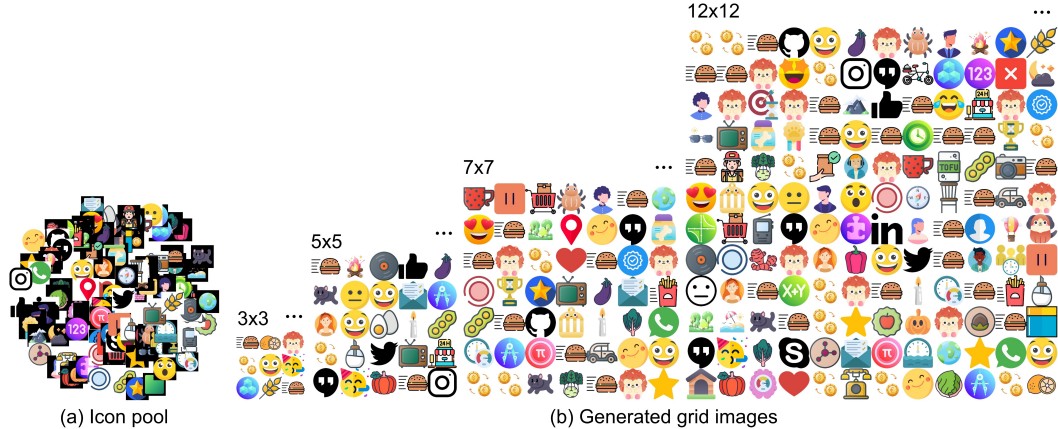

(a) Icon pool             (b) Generated grid images

Figure 8: Visualization of grid images with different icons in content understanding task.

- **Scrambled Text Understanding (STU):** Humans can grasp the general meaning of sentences and paragraphs even with some misspelled words, without significantly affecting overall comprehension. However, whether LLMs can achieve the same remains an open question. To create the questions for this task, we instruct the interviewer model to generate a text of approximately 100 words (*e.g.*, story, diary, or prose), which is then processed by external functions to apply two types of perturbations: 1) disrupt the character order within words. For each text, we flip a certain proportion ($N_{disp}^{word}/N_{total}^{word}$) of correctly spelled words to misspelled words (White et al., 2025), where $N_{disp}^{word}$ and $N_{total}^{word}$ denote the number of disrupted words and the total number of words, respectively; 2) mask a certain proportion ($N_{masked}^{char}/N_{total}^{char}$) of characters with '*', where $N_{masked}^{char}$ and $N_{total}^{char}$ denote the number of masked characters and the total number of characters, respectively. The resulting text is then sent to the interviewee model for completion and comprehension.

- **String Parsing (SP):** To investigate the discernibility of LLMs in processing long texts, we design a needle-in-a-haystack evaluation task, requiring the LLMs to count the occurrences of a target character within a long string. We perform similar instructions to the STU task for string generation and add the needle '-' to a random position within the string. Then, the string length is incrementally increased based on the interviewee's performance to elevate task difficulty.

- **Arithmetic:** Considering the inherent challenge of quantifying the difficulty of mathematical problems, which is normally associated with factors such as problem complexity, level of abstraction, the number of solution steps, and the methods employed to solve it, we select two basic arithmetic problems with quantifiable difficulties: 1) multiplication of decimals with increasing digits, and 2) operations with increasing number of terms.

- **Binary Tree Traversal (BTT):** Binary tree is a common data structure widely used in computer science and programming. In general, a binary tree structure can be uniquely determined by given its pre-order and in-order traversals. Based on this, we prompt the interviewer model to generate a binary tree with its pre-order and in-order traversals, then query the interviewee model to output the post-order traversal. The number of nodes is used to control the task difficulty, namely the structural complexity of a binary tree.

- **Shortest Path Search (SPS):** The shortest path algorithm is an important concept in graph theory used to find the shortest path between two nodes in a weighted graph. Taking the famous Dijkstra algorithm as an example, its time complexity is $O(V^2)$ and $O((V + E)\log V)$ when using a regular array and a priority queue, respectively. Inspired by this, we first instruct the interviewer model to generate a graph with an adjacency matrix along with the start and end nodes, which is further sent to the interviewee model for shortest path computation. The number of nodes and edge density are used to control the difficulty.

- **Code Generation:** Similar to the mathematical problems, the difficulty of coding problems is also hard to define. In this paper, we focus on the cyclomatic complexity, which is a

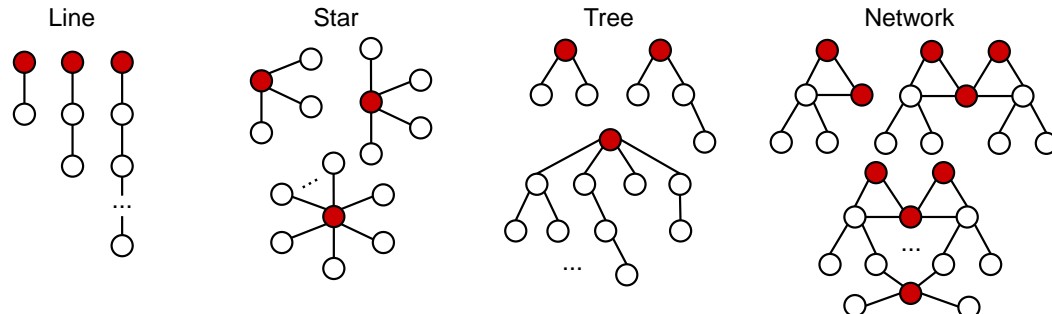

Figure 9: Example of four typical evaluation topologies as well as their variants.

software metric that quantifies the complexity of a code's control flow, specifically measuring the number of linearly independent paths through the code. We first instruct the interviewer model to generate an initial code demand and then send this to the interviewee model for code generation. After that, the generated code will be sent back to the interviewer model for examination. If the interviewee successfully completes the current task, a new requirement will be randomly added, such as incorporating a 'for' loop, a 'while' loop, or an 'if' conditional statement. This process will continue iteratively until the interviewee fails to produce the correct output.

- **Code Debugging:** As for the debugging task, we adopt the same code generation demand to the interviewer model and then mask or delete a certain proportion of them by an external function. The interviewee model is asked to repair the missing sections.

Note that for arithmetic, BTT, and SPS tasks, we implement external code-based supervision to prevent potential generation errors made by the interviewer itself.

## C  ADDITIONAL INFORMATION OF EVALUATION TOPOLOGY

We distinguish evaluation topologies into four primary categories: line, star, tree, and hybrid network, as shown in Fig. 9.

- **Line:** Line-based evaluation topology is the simplest structure where nodes (such as interviewee models and interviewer models) are connected in a sequential manner along a single communication line. In other words, in each round of task evaluation, only one interviewer model engages in dialogue-based assessment with the interviewee model.

- **Star:** Star topology means that one interviewee model is connected to multiple interviewer models, with a maximum hop count of 1. The typical assessment scenarios corresponding to this evaluation topology include: a) $N$ interviewers jointly assess the performance under a specific task and conduct a comprehensive performance discussion or merely average the score; b) for each interviewee, all tasks or capabilities are evaluated in parallel. Each line within a star topology denotes a single evaluation process.

- **Tree:** Tree-based evaluation topology possesses an innate hierarchical structure that enables decomposable evaluations from low-level abilities to high-level abilities. For example, when evaluating the math capability, one can start with the fundamental arithmetic tasks and proceed to more complex calculus, equation solving, and mathematical proof tasks.

- **Hybrid Network:** Compared to other single-type topologies, the hybrid network presents a more intricate web of node relationships, where a new edge type (*interviewee-interviewee*) and a cyclic structure are introduced. Specifically, such a structure is more suited to a panel discussion form of evaluation task. For example, evaluating the collaborative communication skills and debating abilities of large models for more truthful answers, when confronted with the task of designing solutions to complex problems (Khan et al., 2024).

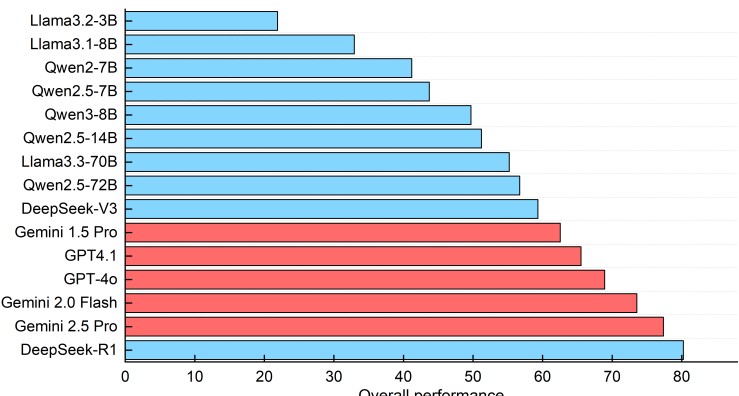

Figure 10: Overall performance comparison among 15 LLMs based on the evaluation network.

Table 5: The SRCC between the score of the model and all raters for 'scoring' setting. The human agreement percentage in the form of experts/crowds for 'yes-or-no' setting.

| Setting | Supervisor | $STU_{disrupt.}$ | $STU_{mask.}$ | $Arith._{oper.}$ | BTT |
|---|---|---|---|---|---|
| Scoring | Claude 3.7 Sonnet | 0.8419 | 0.8593 | 0.8862 | 0.8122 |
| Scoring | Gemini 2.5 Pro | 0.8522 | 0.8668 | 0.8912 | 0.8217 |
| Yes-or-No | Claude 3.7 Sonnet | 0.944/0.923 | 0.935/0.920 | 0.936/0.918 | 0.903/0.898 |
| Yes-or-No | Gemini 2.5 Pro | 0.948/0.929 | 0.938/0.922 | 0.941/0.927 | 0.925/0.908 |

# D  ADDITIONAL EXPERIMENTAL RESULTS

**Subjective Alignment of the Supervisor.** We conducted extra subjective experiments where we quantified their alignment with human annotations. Specifically, we adopt two schemes: 'yes-or-no' evaluation and scoring evaluation (Chen et al., 2024a). 2 experts (with experience in publishing articles on large models) and 8 public participants from campus are invited to participate. We selected four tasks with high error rates (according to Tab. 3), including $STU_{disrupt.}$, $STU_{mask.}$, $Arith._{oper.}$, and BTT, for evaluation. We use Claude 3.7 Sonnet and Gemini 2.5 Pro as the supervisor. GPT-4o is the interviewer. Note that in these tasks, we have already implemented external algorithmic function checks for the ground-truth errors (e.g., "23445 + 22784 = ?"). Therefore, human and third-party supervisory models primarily focus on errors related to textual hallucinations. In 'yes-or-no' evaluation, the human participants and the supervisor model are instructed to answer "*whether there exist any hallucination in the evaluation dialogues? Return yes or no.*" In the scoring evaluation, we instruct "*Please carefully judge the quality of the evaluation process dialogue based on the task requirements*," and provide a detailed scoring standard with a 1-5 Likert scale:

- 1: The response is accurate and trustworthy
- 2: The response is mostly correct but contains minor, non-core factual errors
- 3: A mix of fact and fabrication makes the response partially unreliable
- 4: The core claim is false or the response is mostly fabricated and misleading
- 5: The response is almost entirely invented, nonsensical, or detached from reality.

In Tab. 5, we observe a significant high correlation between human and the supervisor models in both settings, demonstrating their effectiveness in supervising the evaluation process. In addition, we found that over 95% of the queries received a rating of 4 or higher, showing the effectiveness of the joint generation scheme of the interviewer model and the external function.

**Multiple Roles Exploration.** To evaluate the effect of different role settings, we further conduct experiments for the same model in interviewee, interviewer, and supervisor. Concretely, we test one open-source model Qwen2.5-7B and one proprietary model GPT-4o in three representative tasks, including $STU_{disrupt.}$, $Arith._{scale}$, and BTT. As shown in Tab. 9, we can observe that changing the supervisor does not explicitly affect the resulting performance. Meanwhile, after changing the

Table 6: Results on different role settings. ACC-AUC/Max. Perf. is reported.

| Interviewee | Interviewer | Supervisor | $STU_{disrupt.}$ | $Arith._{scale}$ | BTT |
|---|---|---|---|---|---|
| Qwen2.5-7B | GPT-4o | Claude 3.7 Sonnet | 6.797/8 | 0.9/1 | 1.6/2 |
| Qwen2.5-7B | Qwen2.5-7B | Qwen2.5-7B | 6.833/8 | 1.0/2 | 1.6/2 |
| Qwen2.5-7B | Qwen2.5-7B+GPT-4o+Claude 3.7 Sonnet | Qwen2.5-7B | 6.802/8 | 0.9/1 | 1.6/2 |
| GPT-4o | GPT-4o | Claude 3.7 Sonnet | 8.855/10 | 1.7/2 | 3.1/4 |
| GPT-4o | GPT-4o | GPT-4o | 8.855/10 | 1.7/2 | 3.1/4 |

Table 7: Results on different role settings. ACC-AUC/Max. Perf. is reported.

| Interviewee | Interviewer | Supervisor | $STU_{disrupt.}$ | $Arith._{scale}$ | BTT |
|---|---|---|---|---|---|
| Qwen2.5-7B | GPT-4o | Claude 3.7 Sonnet | 6.797/8 | 0.9/1 | 1.6/2 |
| Qwen2.5-7B | Qwen2.5-7B | Qwen2.5-7B | 6.833/8 | 1.0/2 | 1.6/2 |
| Qwen2.5-7B | Qwen2.5-7B+GPT-4o+Claude 3.7 Sonnet | Qwen2.5-7B | 6.802/8 | 0.9/1 | 1.6/2 |
| GPT-4o | GPT-4o | Claude 3.7 Sonnet | 8.855/10 | 1.7/2 | 3.1/4 |
| GPT-4o | GPT-4o | GPT-4o | 8.855/10 | 1.7/2 | 3.1/4 |

interviewer from GPT-4o to Qwen2.5-7B, the evaluated performance of Qwen2.5-7B on $STU_{disrupt.}$ and $Arith._{scale}$ tasks grows slightly. We suspect that this is due to the limited diversity of the generated queries stemming from the relatively weak capability of the interviewer model (similar conclusion as Tab. 4). Therefore, we further employ a 1v3 star evaluation topology to solve the mild variance for the same interviewee model across interviewers, which indeed improves the reliability of the evaluation process. Moreover, we find a high error rate (56%) in the BTT task, calculated by the misalignment between the external function and self-generated ground-truth, when Qwen2.5-7B serves as the interviewer. This result is much lower in GPT-4o's setting, showing the necessity of using relatively powerful models and incorporating external functions.

**Compare ACC-AUC to Static Evaluation Baselines with Difficulty Settings.** We conduct additional experiments for comparing performance trends across easy, medium, and hard tasks in static benchmarks under the framework of MACEval. Specifically, we use the problems from E2H-AMC, a subset in Easy2Hard-Bench (Ding et al., 2024), where the difficulty rating is represented by the percentage of students who answered each question correctly. Here are three problem examples:

- *Cagney can frost a cup cake every <20> seconds and Lacey can frost a cupcake every <30> seconds. Working together, how many cupcakes can they frost in <5> minutes? Avg. Difficulty = 0.134* Avg. Difficulty = 0.134

- *Find the number of pairs of integers (a, b) with $<1> \leqslant a < b \leqslant <57>$ such that $a^2$ has a smaller remainder than $b^2$ when divided by <57>.* Avg. Difficulty = 0.587

- *In a <16>x<16> table of integers, each row and column contains at most <4> distinct integers. What is the maximum number of distinct integers that there can be in the whole table?* Avg. Difficulty = 0.784

We uniformly sample 10 question templates with incremental difficulty rating from 0.1035 to 0.8548 (The larger the value, the higher the difficulty). During the evaluation data generation, the interviewer model with external function-assisted is instructed to generate query-answer by changing the values in '<>'. We set the number of queries at each difficulty level to 100. Rather than relying on manually annotated fine-grained difficulty scores, we increase the number of testing rounds within each predefined difficulty level to obtain a more precise performance estimate. Here, we tested two models: Qwen 2.5-VL-72B and Gemini 1.5 Pro. GPT-4o is used as the interviewer model. In Tab. 8, we observe that the overall declining trend on both dynamic and static settings are similar. The performance under MACEval with multi-round queries exhibit slightly higher values than those with static settings, showing a more genuine performance while demonstrating its potential superiority for longitudinal evaluation against data contamination.

**The Stability of Evaluation Data Generation.** We compare the performance differences in terms of the number of generated queries at the same difficulty level on $STU_{mask.}$ and BTT tasks, as shown in Tab. 10. The interviewee and interviewer are Qwen2.5-72B and GPT-4o, respectively. We further conducted two-side t-test and reported 95% PI and p-value for 300 queries in Tab. 11. We can observe that there is no significant difference between the results of 10@10 and 20@10 that the

Table 8: Data points in discrete form. The upper and bottom parts are the accuracy results under MACEval and static settings, respectively.

| Difficulty level | 1 | 2 | 3 | 4 | 5 | 6 | 7 | 8 | 9 | 10 |
|---|---|---|---|---|---|---|---|---|---|---|
| Qwen2.5-72B | 0.57 | 0.42 | 0.32 | 0.27 | 0.24 | 0.18 | 0.14 | 0.10 | 0.10 | 0.09 |
| Gemini 1.5 Pro | 0.51 | 0.36 | 0.24 | 0.21 | 0.16 | 0.14 | 0.10 | 0.08 | 0.08 | 0.08 |
| Qwen2.5-72B | 0.53 | 0.43 | 0.29 | 0.21 | 0.18 | 0.17 | 0.12 | 0.10 | 0.08 | 0.09 |
| Gemini 1.5 Pro | 0.46 | 0.39 | 0.22 | 0.12 | 0.10 | 0.07 | 0.06 | 0.04 | 0.04 | 0.03 |

Table 9: Results on different role settings. ACC-AUC/Max. Perf. is reported.

| Interviewee | Interviewer | Supervisor | $STU_{disrupt.}$ | $Arith_{\cdot scale}$ | BTT |
|---|---|---|---|---|---|
| Qwen2.5-7B | GPT-4o | Claude 3.7 Sonnet | 6.797/8 | 0.9/1 | 1.6/2 |
| Qwen2.5-7B | Qwen2.5-7B | Qwen2.5-7B | 6.833/8 | 1.0/2 | 1.6/2 |
| Qwen2.5-7B | Qwen2.5-7B+GPT-4o+Claude 3.7 Sonnet | Qwen2.5-7B | 6.802/8 | 0.9/1 | 1.6/2 |
| GPT-4o | GPT-4o | Claude 3.7 Sonnet | 8.855/10 | 1.7/2 | 3.1/4 |
| GPT-4o | GPT-4o | GPT-4o | 8.855/10 | 1.7/2 | 3.1/4 |

p-value>0.05. Their 95% PIs have a considerable overlap, indicating the robustness of the evaluation process. Furthermore, the 95% PI of 20@10 is is more concentrated than that of 10@10, showing the precision of increasing the number of rounds under the given difficulty level.

**Expanding the Task Suite to Include Tasks that are Less Structured and Quantifiable.** We further expanded the task suite to include language understanding tasks that are less structured or quantifiable. Specifically, we focus on the following tasks:

- External material: To avoid data contamination, we manually collected 10 recent news from world economic forum[3] (time period: 2025.7.20-7.25) to ensure that they are not in the training data.
- Self-generated: The interviewee model is required to generate text contents with certain length and requirements.

Settings:

- For task 1-1 (External), where the interviewer randomly require the interviewee model to paraphrase, simplify, or summarize the given text content with certain output format, we score tasks purely by their adherence to the instructions (White et al., 2025). The interviewer model are instructed to increase the number of queries in each round until the finishing rate equal to a very low value. However, in this experiments, we found a relatively slow rate of decline. It is time-consuming to reach the scenario with 0% finishing rate compared to other tasks. Therefore, we report the ACC-AUC at 50% finishing rate and the corresponding number of queries, which can also validate the feasibility of MACEval in handling open language understanding tasks.

- For task 1-2 (External), we further employ the language bootstrapping (Yang et al., 2024c) (e.g., the proportion of irrelevant words or sentences) to adjust the difficulty rate each success round while satisifying the principle of MACEval (pushing to the limit). Therefore, we report the ACC-AUC and the bootstrapping proportion (%).

- For task 2 (Self-generated), the interviewer model are instructed to increase the number of queries (difficulty rate) in each round until the finishing rate equal to a very low value. A external Python function is deployed to count word length for accuracy calculation. Same with scenario 1-1, we report the ACC-AUC at 50% finishing rate and the corresponding number of queries due to the slow decline rate.

- Four models are selected for evaluation. Three different evaluation topologies are investigated (including 1-hop line and 1v4 star).

In Tab. 12, we find tiny difference in ACC-AUC/Max. Perf. when changing GPT-4o to Claude 3.7 Sonnet under 1-hop line. This can be attributed to the small disparity of the in-process data, which

---

[3]https://www.weforum.org/

Table 10: Results on different numbers of queries. ACC-AUC/Max. Perf. is reported.

| #Query | $STU_{mask.}$ | BTT |
|--------|---------------|--------|
| 10 | 4.247/5 | 2.40/3 |
| 20 | 4.256/5 | 2.50/3 |
| 30 | 4.253/5 | 2.43/3 |
| 50 | 4.244/5 | 2.46/3 |

Table 11: Results of the statistical test.

| #Query@N times | $STU_{mask.}$ | BTT |
|----------------|---------------|--------|
| 10@10 | [4.2444, 4.2558] | [2.4122, 2.4618] |
| 20@10 | [4.2457, 4.2533] | [2.4265, 2.4570] |
| p-value | 0.8486 | 0.7227 |

can be mitigated under the 1v4 star evaluation topology, where a mean opinion score (MOS)-based strategy is adopted, showing good robustness and flexibility of the MACEval framework. We notice the results in Task 1-1 and Task-2 are similar, indicating that the influence of the data source in certain tasks is negligible and validating the suitability of AI-generated text in such tasks.

**More Evaluation on Math Tasks.** We expand the math tasks to solving the equations, we choose 8 models for evaluation, where GPT-4o serves as the interviewer model. The difficulty level is set to the number of variables. We require it to retain four decimal places. As listed in Tab. 13, we observe that those open-source models even with large scale parameters or mainstream proprietary models, such as GPT-4o and Gemini1.5 Pro, can only solve equation with two variables, showing great potential for general large models to improve. Surprisingly, the cutting-edge Gemini 2.5 Pro, a model skilled in mathematics and reasoning, is only capable of solving systems of three linear equations in three variables to a limited extent.

**Potential Error Propagating.** We conduct extra experiments to explore the effects of that errors are proactively injected to the next round under the current node or evaluation process. Specifically, we add interference characters (unordered and meaningless strings) into the message, which are together taken to the next node. We evaluate on the data generation process of the binary tree traversal (BTT) task, which have the highest inital error (52%, Tab. 3). The results in Tab. 14 show that injecting irrelevant textual contents will not affect the inherent error rate that depends on the difficulty of the task itself. Conversely, the progressive accumulation of erroneous content renders the information more readily detectable by third-party supervisor, facilitating its timely suppression and the issuance of a regeneration directive to the interviewer model. Hence, the applied external algorithm played a crucial role within the single task.

## D.1 COMPARISON TO OTHER LARGE MODEL BENCHMARKS

We compare the evaluation results obtained by MACEval to the popular benchmarks (LiveBench (White et al., 2025) and ChatBot Arena (Chiang et al., 2024)), shown in Fig. 11. It can be observed that the results of MACEval highly correlate with mainstream benchmarks (avg. $r = 0.9509$), demonstrating the effectiveness of MACEval and proving the potential of open-ended problem settings.

## D.2 VISUALIZATION OF THE EVALUATION STREAM

To illustrate the working mechanism of MACEval more intuitively, we visualize the dialogue example of the third-party supervisor and prompt templates for both interviewee and interviewer in Fig. 12 and Appendix F. In this paper, we adopt the Claude 3.7 Sonnet (Anthropic, 2024) as the supervisor.

Table 12: Results on the expanded language understanding tasks.

| Model | Interviewer | Task1-1 | Task1-2 | Task2 |
|---|---|---|---|---|
| GPT-4o | | 54.4/88 | 41.3/45 | 57.5/93 |
| Gemini 2.0 Flash | GPT-4o | 67.2/113 | 40.8/44 | 70.3/116 |
| Qwen2.5-72B | | 49.8/75 | 41.9/45 | 51.6/77 |
| Qwen2.5-7B | | 21.5/26 | 29.3/38 | 21.8/27 |
| GPT-4o | | 53.8/87 | 41.9/45 | 58.9/95 |
| Gemini 2.0 Flash | Claude 3.7 Sonnet | 65.8/109 | 42.7/46 | 74.4/119 |
| Qwen2.5-72B | | 50.3/77 | 41.7/45 | 53.4/78 |
| Qwen2.5-7B | | 21.9/26 | 29.5/39 | 22.4/28 |
| GPT-4o | | 55.6/88 | 41.7/45 | 58.2/95 |
| Gemini 2.0 Flash | GPT-4o+Claude 3.7 Sonnet+Gemini 2.5 Pro+DeepSeek V3 | 67.1/112 | 41.7/45 | 73.8/117 |
| Qwen2.5-72B | | 50.8/76 | 41.7/45 | 52.9/77 |
| Qwen2.5-7B | | 21.8/26 | 29.6/39 | 21.9/27 |

Table 13: Results on the equation solving task.

| Model | Equation Solving |
|---|---|
| GPT-4o | 1.4/2 |
| Gemini 2.5 Pro | 2.1/3 |
| Gemini 2.0 Flash | 1.4/2 |
| Gemini 1.5 Pro | 1.1/2 |
| Qwen2.5-7B | 0.8/1 |
| Qwen2.5-14B | 0.8/1 |
| Qwen2.5-72B | 1.3/2 |
| Llama3.1-8B | 0.6/1 |

# E    FUTURE POSSIBILITIES

Apart from the comments for possible next steps of research related to the evaluation of large models that have already been given, this section is devoted to the extension for some of them and then more topics with good potential based upon our understanding and rethinking for the field.

- **Focus on Efficient Evaluation.** Currently, most evaluations are trending towards larger scales and broader scopes, which significantly exacerbate the costs associated with data collection and maintenance. Therefore, the development of efficient evaluation pathways is of great importance. For instance, it may be possible to assess a large model's capabilities in a particular dimension by evaluating a small number of key samples. Conducting principal component analysis and dimensionality reduction on evaluation data to identify efficient and rapid evaluation routes could emerge as a significant research direction.

- **Evaluating the evaluations.** Evaluations are critical for understanding the capabilities of large models. Its fairness, comprehensiveness, reproducibility, timeliness, scalability, transparency, and practicality are key points to serve as a "good" evaluation. Recently, with the rapid increasing number of various benchmarks, it is highly needed for objective metrics to evaluate evaluations themselves.

It is hoped that these can provide actionable ideas for researchers and would trigger further discussion, and more importantly, new exploration in this area.

# F    PROMPT TEMPLATES

Table 14: Impact of error propagation.

| Error-round 1 | Error-round 2 | Error-round 3 | Error-round 4 | Error-round 5 |
|---|---|---|---|---|
| 52% | 50% | 53% | 52% | 54% |

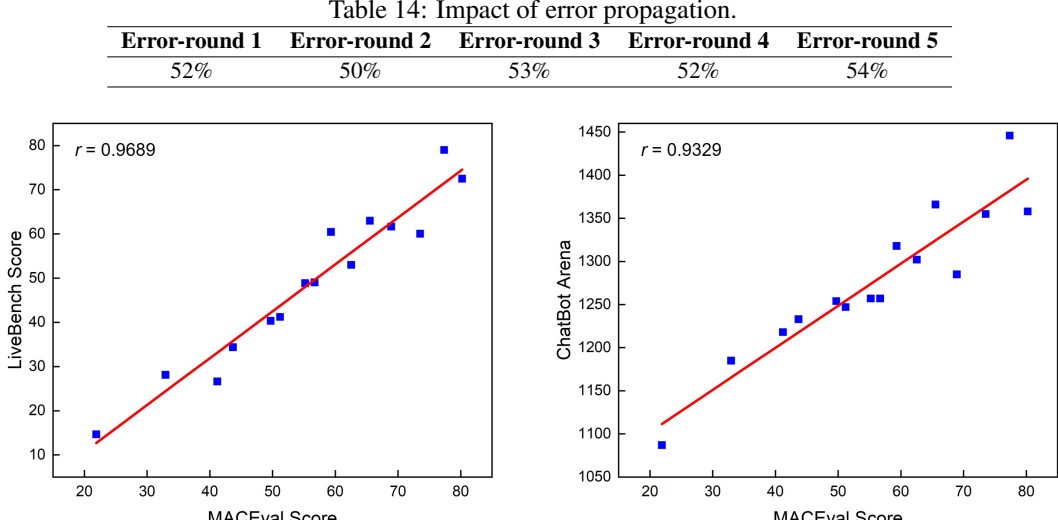

Figure 11: The performance of LLMs on different benchmarks, compared to a best-fit line. We compare the evaluation results in relative performance on MACEval vs. LiveBench, and MACEval vs. ChatBot Arena.

### F.1 PROMPT TEMPLATES FOR VISUAL PERCEPTION TASKS

**Example for Interviewer:**

```
<|System Prompt|>
You are a helpful assistant for prompt generation in text-to-image
tasks.

<|User Prompt|>
Please generate a prompt regarding the generation of {keyword}
images.  Return text in <text></text> format.

<|Interviewer's Answer|>
<text>A high-resolution natural landscape at sunrise, viewed from
a hilltop.  The foreground shows dewy grass and wildflowers gently
swaying in the breeze.  In the middle ground, a winding dirt path
leads through a lush green forest with scattered rays of sunlight
filtering through the trees.  In the background, soft morning mist
blankets a valley with a river glistening under the golden light of
the rising sun.  Birds are flying low near the horizon, and the sky
transitions from deep orange to pale blue.</text>

<|External image generation model|>
def SD3_5 (prompt):

  ......

<|External distortion algorithm|>
def add_noise (img, noise_level):

  ......

<|Resulting content|>
Reference image:  <image1>
Distortion image:  <image2>
Noise level:  <noise_level>
```

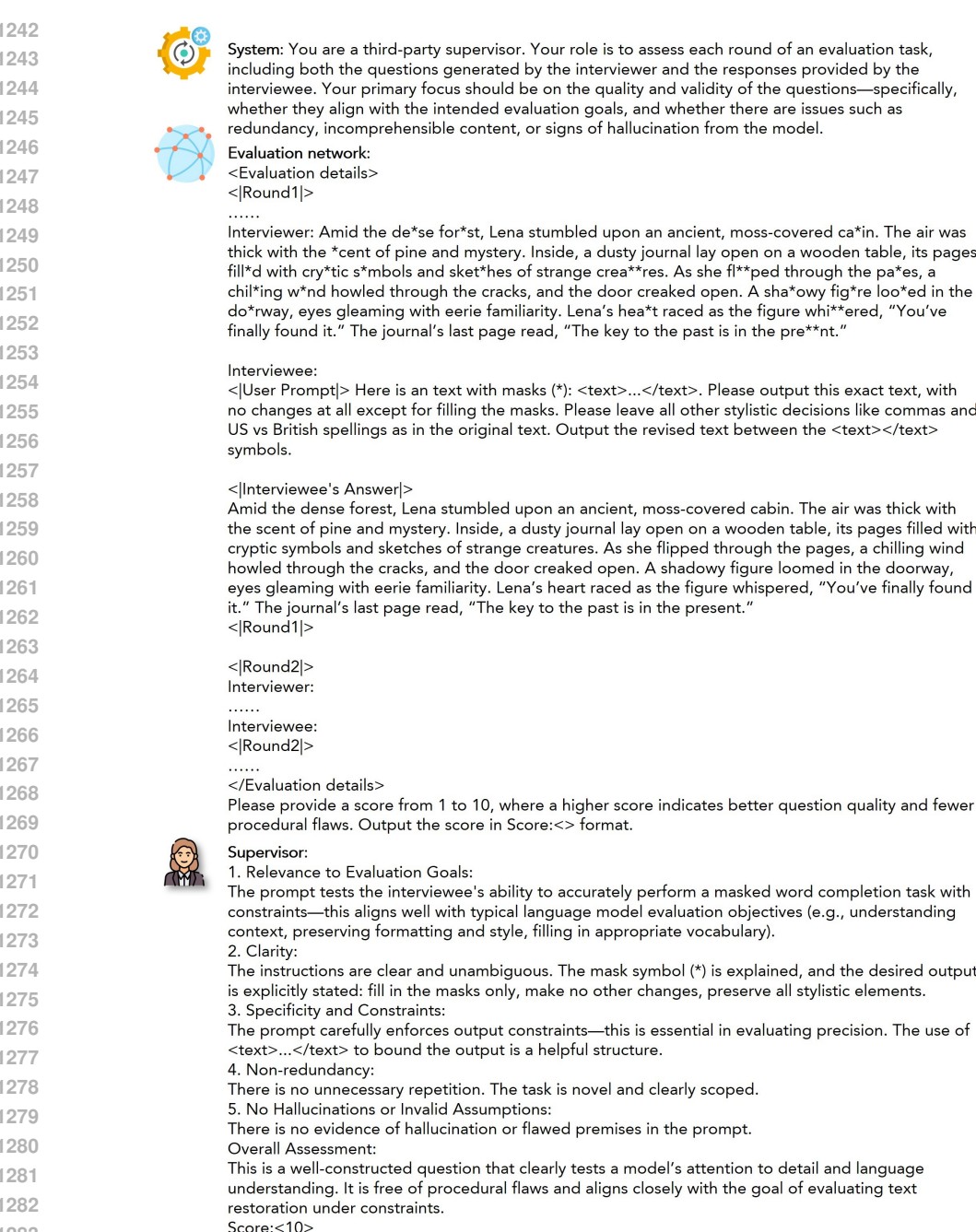

**System:** You are a third-party supervisor. Your role is to assess each round of an evaluation task, including both the questions generated by the interviewer and the responses provided by the interviewee. Your primary focus should be on the quality and validity of the questions—specifically, whether they align with the intended evaluation goals, and whether there are issues such as redundancy, incomprehensible content, or signs of hallucination from the model.

**Evaluation network:**
<Evaluation details>
<|Round1|>
......
Interviewer: Amid the de*se for*st, Lena stumbled upon an ancient, moss-covered ca*in. The air was thick with the *cent of pine and mystery. Inside, a dusty journal lay open on a wooden table, its pages fill*d with cry*tic s*mbols and sket*hes of strange crea**res. As she fl**ped through the pa*es, a chil*ing w*nd howled through the cracks, and the door creaked open. A sha*owy fig*re loo*ed in the do*rway, eyes gleaming with eerie familiarity. Lena's hea*t raced as the figure whi**ered, "You've finally found it." The journal's last page read, "The key to the past is in the pre**nt."

Interviewee:
<|User Prompt|> Here is an text with masks (*): <text>...</text>. Please output this exact text, with no changes at all except for filling the masks. Please leave all other stylistic decisions like commas and US vs British spellings as in the original text. Output the revised text between the <text></text> symbols.

<|Interviewee's Answer|>
Amid the dense forest, Lena stumbled upon an ancient, moss-covered cabin. The air was thick with the scent of pine and mystery. Inside, a dusty journal lay open on a wooden table, its pages filled with cryptic symbols and sketches of strange creatures. As she flipped through the pages, a chilling wind howled through the cracks, and the door creaked open. A shadowy figure loomed in the doorway, eyes gleaming with eerie familiarity. Lena's heart raced as the figure whispered, "You've finally found it." The journal's last page read, "The key to the past is in the present."
<|Round1|>

<|Round2|>
Interviewer:
......
Interviewee:
<|Round2|>
......
</Evaluation details>
Please provide a score from 1 to 10, where a higher score indicates better question quality and fewer procedural flaws. Output the score in Score:<> format.

**Supervisor:**
1. Relevance to Evaluation Goals:
The prompt tests the interviewee's ability to accurately perform a masked word completion task with constraints—this aligns well with typical language model evaluation objectives (e.g., understanding context, preserving formatting and style, filling in appropriate vocabulary).
2. Clarity:
The instructions are clear and unambiguous. The mask symbol (*) is explained, and the desired output is explicitly stated: fill in the masks only, make no other changes, preserve all stylistic elements.
3. Specificity and Constraints:
The prompt carefully enforces output constraints—this is essential in evaluating precision. The use of <text>...</text> to bound the output is a helpful structure.
4. Non-redundancy:
There is no unnecessary repetition. The task is novel and clearly scoped.
5. No Hallucinations or Invalid Assumptions:
There is no evidence of hallucination or flawed premises in the prompt.
Overall Assessment:
This is a well-constructed question that clearly tests a model's attention to detail and language understanding. It is free of procedural flaws and aligns closely with the goal of evaluating text restoration under constraints.
Score:<10>

Figure 12: The working mechanism of the third-party supervisor involved along the evaluation pathway and responsible for validating the entire evaluation network.

**Example for Interviewee:**

```
<|User Prompt|>
Given two images <image1> and <image2>, please answer if there are
any differences in noise between them.  Return judgment <yes> or
<no> with analysis in format:
<Judgment>xxx</Judgment>
<Analysis>xxx</Analysis>

<|Interviewee's Answer|>
<Judgment>No</Judgment>
<Analysis>There are no differences between these two images.  They
appear to be identical.</Analysis>
```

## F.2   PROMPT TEMPLATES FOR CONTENT UNDERSTANDING TASKS

**Example for Interviewer:**

```
<|System Prompt|>
You are a helpful assistant.

<|User Prompt|>
Please generate two random numbers from 1 to 100 and from 1 to
{{level+2} × {level+2}}, respectively.  Return answer in the
following format:
Number1:  <> # The first one denotes the selected icon's ID.
Number2:  <> # The second one denotes the number of the same icons.

<|Interviewer's Answer|>
Number1:  <67>
Number2:  <11>

<|External grid_image generation algorithm|>
def grid_image (image_id, size):

   ......

<|Resulting content|>
Icon image:  <image1>
Grid image:  <image2>
Grid size:  <level+2>
```

**Example for Interviewee:**

```
<|User Prompt|>
Given an icon image <image1> and a large grid image <image2>,
please answer how many icons <image1> are there in the grid image
<image2>?  Return number only in format:
Number:  <>

<|Interviewee's Answer|>
Number:  <4>
```

## F.3   PROMPT TEMPLATES FOR SCRAMBLED TEXT UNDERSTANDING TASKS

**Example for Interviewer (Disruption):**

```
<|System Prompt|>
You are an expert in the field of literature, specializing in
writing stories, journals, and essays.

<|User Prompt|>
Please generate a piece of text, about 100 words, which can be a
story, journal, or essay.  Output the text between the <text></text>
symbols.

<|Interviewer's Answer|>
<text>
Amid the dense forest, Lena stumbled upon an ancient, moss-covered
cabin.  The air was thick with the scent of pine and mystery.
Inside, a dusty journal lay open on a wooden table, its pages
filled with cryptic symbols and sketches of strange creatures.  As
she flipped through the pages, a chilling wind howled through the
cracks, and the door creaked open.  A shadowy figure loomed in the
doorway, eyes gleaming with eerie familiarity.  Lena's heart raced
as the figure whispered, "You've finally found it." The journal's
last page read, "The key to the past is in the present."
</text>

<|Adding Perturbations (Disrupt the character order)|>
def disrupt_words (ori_text, disp_ratio):

  ......

<|Resulting Text|>
Amid the dnese foerst, Lena stubmled upon an nacient, moss-covered
cabin.  The air was tihck with the scent of pnie and msytery.
Inside, a dusty jorunal lay open on a wooedn table, its paegs
filled with cryptic syombls and skecthes of starnge crateures.  As
she flipped thurogh the pages, a chilling wnid howled through the
cracks, and the door crekaed open.  A sahdowy fiugre loomed in the
doroway, eyes gleaming with eerie famiilarity.  Lena's hraet raced
as the fiugre whispered, "You've fainlly found it." The journal's
last page read, "The key to the past is in the present."
```

**Example for Interviewee (Disruption):**

**<|User Prompt|>**
Here is an text with typos: <text>...</text>. Please output
this exact text, with no changes at all except for fixing the
misspellings. Please leave all other stylistic decisions like
commas and US vs British spellings as in the original text. Output
the revised text between the <text></text> symbols.

**<|Interviewee's Answer|>**
<text>
Amid the dense forest, Lena stumbled upon an ancient, moss-covered
cabin. The air was thick with the scent of pine and mystery.
Inside, a dusty journal lay open on a wooden table, its pages
filled with cryptic symbols and sketches of strange creatures. As
she flipped through the pages, a chilling wind howled through the
cracks, and the door creaked open. A shadowy figure loomed in the
doorway, eyes gleaming with eerie familiarity. Lena's heart raced
as the figure whispered, "You've finally found it." The journal's
last page read, "The key to the past is in the present."
</text>

**Example for Interviewer (Mask):**

```
<|System Prompt|>
same as the above

<|User Prompt|>
same as the above

<|Interviewer's Answer|>
same as the above

<|Adding Perturbations (Mask characters with * )|>
def mask_characters (ori_text, mask_ratio):

  ......

<|Resulting Text|>
Amid the de*se fo**st, Lena stu**led up*n an an**ent, moss-cov*red
cabin.  The a*r was thi*k with the scent of p*ne and mys*ery.
In*ide, a *usty jour*al lay op*n on a **oden ta*le, its pa*es
filled with cryptic s**bols and sket**es of st**nge cre*tu*es.  As
she fl**ped through the p*ges, a chilling w**d howled thro*gh the
c*acks, and the door creaked open.  A sh***wy f*gure loomed in the
d**rway, eyes gleaming with eerie famil**rity.  Lena's h*art raced
as the fi**re whis*ered, "You've fina**y found it." The jou**al's
last page read, "The key to the past is in the pr*sent."
```

**Example for Interviewee (Mask):**

```
<|User Prompt|>
Here is an text with masks (*):  <text>...</text>.  Please output
this exact text, with no changes at all except for filling the
masks.  Please leave all other stylistic decisions like commas
and US vs British spellings as in the original text.  Output the
revised text between the <text></text> symbols.

<|Interviewee's Answer|>
<text>
Amid the dense forest, Lena stumbled upon an ancient, moss-covered
cabin.  The air was thick with the scent of pine and mystery.
Inside, a dusty journal lay open on a wooden table, its pages
filled with cryptic symbols and sketches of strange creatures.  As
she flipped through the pages, a chilling wind howled through the
cracks, and the door creaked open.  A shadowy figure loomed in the
doorway, eyes gleaming with eerie familiarity.  Lena's heart raced
as the figure whispered, "You've finally found it." The journal's
last page read, "The key to the past is in the present."
</text>
```

## F.4 PROMPT TEMPLATES FOR STRING PARSING TASKS

**Example for Interviewer:**

```
<|System Prompt|>
You are a helpful assistant proficient in generating text.

<|User Prompt|>
Please generate a {level}-character-long string that may include
English letters (both uppercase and lowercase) and special
characters such as:  !, @, #, %, &.  Then, insert four '-'
characters at random positions in the string.  Return string in
String:  <> format.

<|Interviewer's Answer|>
String:  <a-D#fG%kL-qW!zXe@R-tY&>

<|External algorithm supervision|>
def task_calibration (ori_text):

  ......

<|Resulting Text|>
String:<a-D#fG%kL-q-W!zXe@R-tY&>
```

**Example for Interviewee:**

```
<|User Prompt|>
Here is a long string:  <text>...</text>.  How many '-' characters
are there in this string?  Return the number in the following
format:
Number:  < >

<|Interviewer's Answer|>
Number:  <4>
```

## F.5   PROMPT TEMPLATES FOR ARITHMETIC TASKS

**Example for Interviewer (Scale):**

```
<|System Prompt|>
You are a helpful assistant proficient in mathematics.

<|User Prompt|>
Please generate a multiplication problem with the correct answer
involving two n-digit decimal numbers.  The two numbers must be
different from those used in previous problems.

Return question and correct answer in the following format:
<question> </question>
<answer> <answer>

<|Interviewer's Answer|>
<question> What is 123.456 × 789.123?  </question>
<answer> 97406.100088 </answer>

<|External algorithm supervision|>
def task_calibration (ori_text):

  ......

<|Resulting Text|>
<question> What is 123.456 × 789.123?  </question>
<answer> 97421.969088 </answer>
```

**Example for Interviewee (Scale):**

```
<|User Prompt|>
Please solve the given math problem and return your answer in
<answer> </answer> format.

<|Interviewer's Answer|>
<answer> 97461.969 </answer>
```

**Example for Interviewer (Operation):**

```
<|System Prompt|>
You are a helpful assistant proficient in mathematics.

<|User Prompt|>
Please generate an arithmetic problem with the following
requirements:
1.  Each number should have exactly {digit} significant digits
2.  The numbers should be randomly generated, not following any
pattern
3.  The problem should use exactly {level+1} operators such as
(+,−,×,/)
4.  You can use parentheses () freely to group operations
5.  The numbers should be different from the former problems

Return question and correct answer in the following format:
<question> </question>
<answer> <answer>

<|Interviewer's Answer|>
<question> (12846 + (90572/43105)) × 76230 − 50418 </question>
<answer> 979566891.3525446 </answer>

<|External algorithm supervision|>
def task_calibration (ori_text):

  ......

<|Resulting Text|>
<question> (12846 + (90572/43105)) × 76230 − 50418 </question>
<answer> 979360336.076325 </answer>
```

**Example for Interviewee (Operation):**

```
<|User Prompt|>
Please solve the given math problem and return your answer in
<answer> </answer> format.

<|Interviewer's Answer|>
<answer> 979360426.076235 </answer>
```

## F.6  PROMPT TEMPLATES FOR BINARY TREE TRAVERSAL TASKS

**Example for Interviewer:**

```
<|System Prompt|>
You are a helpful assistant proficient in computer science
algorithms.

<|User Prompt|>
Please generate a binary tree traversal problem with the following
requirements:
1.  The tree should have exactly {level+2} levels
2.  The number of nodes should be between {2**(level+1)} and
{2**(level+2)-1}(inclusive)
3.  The sequences should be different from these last problems:
{last problems}
4.  The tree structure should be a valid binary tree
5.  Each node ID should be a unique integer

Provide three types of traversal sequences.  Format your response
as:
<preorder> </preorder>
<inorder> </inorder>
<postorder> </postorder>

<|Interviewer's Answer|>
<preorder> 50 30 20 10 15 13 12 40 35 70 60 80 75 78 76 79 77
</preorder>
<inorder> 12 13 15 10 20 30 35 40 50 60 70 75 76 78 77 79 80
</inorder>
<postorder> 12 13 15 10 20 35 40 30 60 76 77 79 78 75 80 70 50
</postorder>

<|External algorithm supervision|>
def task_calibration (ori_text):

  ......

<|Resulting Text|>
<preorder> 50 30 20 10 15 13 12 40 35 70 60 80 75 78 76 79 77
</preorder>
<inorder> 12 13 15 10 20 30 35 40 50 60 70 75 76 78 77 79 80
</inorder>
<postorder> 12 13 15 10 20 35 40 30 60 76 77 79 78 75 80 70 50
</postorder>
```

**Example for Interviewee:**

```
<|User Prompt|>
Given the preorder and inorder traversals of a binary tree (with
node IDs separated by spaces) <preorder> ...  </preorder> and
<inorder> ...  </inorder>, compute its postorder traversal.  Please
solve the given algorithm problem and return your answer in <answer>
</answer> format.

<|Interviewer's Answer|>
<answer>12 13 15 10 20 35 40 30 60 76 77 79 78 75 80 70 50</answer>
```

## F.7  PROMPT TEMPLATES FOR SHORTEST PATH SEARCH TASKS

**Example for Interviewer:**

**<|System Prompt|>**
You are a helpful assistant proficient in computer science algorithms.

**<|User Prompt|>**
Please generate a shortest path search problem with the following requirements:
1.  The network should be an undirected weighted graph
2.  The network is given in an adjacency list format.  For example, for each node, its connected nodes along with the corresponding edge weights are listed as:
<network>
A: B(2), C(4)
B: A(2), D(3)
C: A(4), D(1)
D: B(3), C(1)
</network>
3.  The number of nodes is {level_node+4}, or the average edge number of each node is {level_edge+2}
5.  Node IDs must be uppercase or lowercase English letters.

Please randomly select two nodes from the graph and compute the weighted shortest path between them (return the number only).
Format your response as:
<Node ID1> </Node ID1>
<Node ID2> </Node ID2>
<path> </path>

**<|Interviewer's Answer|>**
<network>
A: B(3), C(1), F(8)
B: A(3), D(5), E(7)
C: A(1), D(2)
D: B(5), C(2), E(1)
E: B(7), D(1), F(4)
F: A(8), E(4)
</network>
<Node ID1> A </Node ID1>
<Node ID2> E </Node ID2>
<path> 4 </path>

**<|External algorithm supervision|>**
def task_calibration (ori_text):

    ......

**<|Resulting Text|>**
<network>
A: B(3), C(1), F(8)
B: A(3), D(5), E(7)
C: A(1), D(2)
D: B(5), C(2), E(1)
E: B(7), D(1), F(4)
F: A(8), E(4)
</network>
<Node ID1> A </Node ID1>
<Node ID2> E </Node ID2>
<path> 4 </path>

**Example for Interviewee:**

```
<|User Prompt|>
Given the nodes <Node ID1> A </Node ID1> and <Node ID2> E </Node
ID2> of a undirected weighted graph <network> </network>.
The network is given in an adjacency list format.  For example, for
each node, its connected nodes along with the corresponding edge
weights are listed as:
<network>
A: B(2), C(4)
B: A(2), D(3)
C: A(4), D(1)
D: B(3), C(1)
</network>
Please calculate the weighted shortest path between them and return
your answer in <answer> </answer> format with the number only.

<|Interviewer's Answer|>
<answer> 4 </answer>
```

## F.8 PROMPT TEMPLATES FOR CODE GENERATION TASKS

**Example for Interviewer:**

```
<|System Prompt|>
You are a helpful programming assistant and an expert Python
programmer.

<|User Prompt|>
Please generate a common coding problem specification without any
code and the test input as well as the corresponding printable
output in the following format:
<problem> </problem>
<input> </input>
<output> </output>
Note that the new problem should be different from the {former
problems}.

<|Interviewer's Answer|>
<problem> Given a list of integers, determine whether any two
distinct numbers in the list add up to a specific target number.
Return 'True' if such a pair exists, otherwise return 'False'.
</problem>
<input> numbers = [2, 7, 11, 15], target = 9 </input>
<output> True </output>
```

**Example for Interviewee:**

```
<|User Prompt|>
Given a coding question and the test input as well as the
corresponding printable output in the following format:
<problem> </problem>
<input> </input>
<output> </output>
Please generate a correct Python program that matches the
specification and output requirements.  The code must be directly
executable, so please import the necessary library files and pay
attention to code indentation and line breaks.  Return the code for
two parts in the following format:
<code_function> </code_function>
if __name__ == `__main__':
<code_main> </code_main>

<|Interviewee's Answer|>
<code_function>
from typing import List
def has_pair_with_sum(numbers:  List[int], target:  int) -> bool:
seen = set()
for num in numbers:
complement = target - num
if complement in seen:
return True
seen.add(num)
return False
</code_function>
<code_main>
numbers = [2, 7, 11, 15]
target = 9
print(has_pair_with_sum(numbers, target))
</code_main>
```

## F.9 PROMPT TEMPLATES FOR CODE DEBUGGING TASKS

**Example for Interviewer:**

**<|System Prompt|>**
You are a helpful programming assistant and an expert Python
programmer.

**<|User Prompt|>**
First, please generate a common coding problem specification and
the test input as well as the corresponding printable output in the
following format:
<problem> </problem>
<input> </input>
<output> </output>
Then, generate a correct Python program that matches the
specification and output requirements. The code must be directly
executable, so please import the necessary library files and pay
attention to code indentation and line breaks. Return the code for
two parts in the following format:
<code_function> </code_function>
if __name__ == '__main__':
<code_main> </code_main>
Note that the new problem should be different from the {former
problems}.

**<|Interviewer's Answer|>**
<problem> Given a list of integers, determine whether any two
distinct numbers in the list add up to a specific target number.
Return 'True' if such a pair exists, otherwise return 'False'.
</problem>
<input> numbers = [2, 7, 11, 15], target = 9 </input>
<output> True </output>
<code_function>
```
from typing import List
def has_pair_with_sum(numbers:  List[int], target:  int) -> bool:
seen = set()
for num in numbers:
complement = target - num
if complement in seen:
return True
seen.add(num)
return False
```
</code_function>
<code_main>
```
numbers = [2, 7, 11, 15]
target = 9
print(has_pair_with_sum(numbers, target))
```
</code_main>

**<|External code masking|>**
```
def code_masking (code, masking_ratio):

  ......
```

**<|Resulting Content|>**
<code_function>
```
from ty***g im**rt List
def has_pair_with_sum(numbers:  List[int], target:  int) -> bool:
seen = set()
for num in numbers:
comp***ent = t**get - num
if comp**ment in seen:
return True
seen.******
return False
```
</code_function>

**Example for Interviewee:**

```
<|User Prompt|>
Given a coding question and the test input as well as the
corresponding printable output in the following format:
<problem> </problem>
<input> </input>
<output> </output>
Please repair the masked Python code <code_function>
</code_function> to match the specification and output requirements.
The code must be directly executable, so please import the
necessary library files and pay attention to code indentation and
line breaks.  Then generate a fixed version of the program in the
following format:
<code_function> </code_function>

<|Interviewee's Answer|>
<code_function>
from typing import List
def has_pair_with_sum(numbers:  List[int], target:  int) -> bool:
seen = set()
for num in numbers:
complement = target - num
if complement in seen:
return True
seen.add(num)
return False
</code_function>
```

