# OpenReview forum: "MACEval: A Multi-Agent Continual Evaluation Network for Large Models"
_ICLR.cc/2026/Conference — ICLR 2026 Conference Withdrawn Submission_

### Official Review · Reviewer_SaU7 · 2025-10-20

**Soundness:** 2
**Presentation:** 2
**Contribution:** 1
**Rating:** 2
**Confidence:** 4

**Summary:**

This paper introduces **MACEval**, an automated multi-agent evaluation framework for large language models (LLMs).  The core idea is to move beyond static benchmarks by letting LLMs themselves propose new problems, generate candidate solutions, and act as judges in a closed-loop evaluation pipeline.  The authors claim that such a *multi-agent, saturation-resistant* benchmark allows continuous discovery of challenging tasks without human intervention.   MACEval reportedly adjusts task difficulty dynamically and provides automatic scoring based on agent interactions.   The paper positions this work as a step toward scalable, self-improving LLM evaluation.

The paper has two central weaknesses.  First, it **does not report** the new and challenging tasks that the pipeline has dynamically generated, so nobody knows really whether those tasks are good or bad.  For a paper submitted to datasets and benchmarks, this problem really hurts.

Second, the tasks, which are not reported, are most likely **low-quality, contrived, pointless, low in construct validity or even wrong**.  The proposed math tasks are so hard that gemini-2.5-pro, according to the authors, cannot solve a system of linear equations with 3 variables.  (Who believes that?). The pipeline makes coding tasks arbitrarily and pointlessly harder by adding a lot more `for` and `while` loops.  (Who cares about such tasks?)

I myself believe that multi-agent eval is the way to go and have experimented similar ideas months ago.  I have found that the core difficulty of doing this is precisely that **LM-generated difficult tasks become pointless and meaningless very quickly**.  This paper does not contain any ideas that I did not try when I ran my experiments, and instead contains many suggestive signs that those synthetically generated tasks are of low-quality, so my best guess is that those tasks are as low-quality as the ones I produced.

For now, my overall recommendation is **rejection**.  While the conceptual direction—automated, multi-agent benchmarking—is correct and timely, the current implementation lacks transparency, reproducibility, and conceptual rigor.  Strong revision with released tasks, code, and clearer construct validity analysis could make this a valuable contribution in future iterations.

**Strengths:**

1. **Motivation and vision.**
   The authors correctly identify a central problem in modern benchmarking: *saturation*.  As top models quickly approach near-perfect scores on traditional datasets, automatic generation of new and harder tasks is a promising direction.  The proposed “multi-agent” perspective—where one LLM proposes tasks, another solves them, and a third judges—captures the spirit of adaptive evaluation.  Conceptually, this is aligned with current interest in *adversarial*, *continual*, and *self-play* evaluation methods.

2. **Relevance.**
   The work addresses a pressing bottleneck: human-authored benchmarks cannot keep pace with the speed of model scaling.  The vision of having autonomous agents design and grade tasks is highly appealing, especially for reasoning, coding, and planning domains.

3. **Potential originality.**
   While similar efforts (e.g., *self-challenging agents* from Meta) exist, MACEval’s multi-agent framing offers a distinctive lens if implemented correctly.   Even a partial success in generating meaningful new tasks could have impact.

**Weaknesses:**

This paper is likely problematic, but due to lack to transparency (point 1), it is not easy to pin down the exact problem.  I can explain my reasoning based on some circumstantial evidence (points 2-4).  My best guess is that this paper makes the same mistake as in *Illusion of Thinking*, i.e., **misunderstanding contrived complication for difficulty with realistic value**.

1. **Lack of transparency in the generated tasks.**
   The most severe weakness is the *absence of released or described tasks*.  Since the paper claims to generate novel and challenging problems that even frontier models fail to solve, the readers must be able to inspect examples.  Without this, the claimed contribution is unverifiable.  For a “datasets and benchmarks” track, this omission is fatal—the contribution of the paper hinges heavily on the quality of generated tasks.

   > **Actionable fix:** Release at least 20 representative generated tasks (including both the prompt and reference solution) across domains.
   > Provide human evaluation of task validity and difficulty.


Because I cannot see the tasks themselves, I really cannot tell what is going on behind the scenes, but there are many telltale signs suggesting that the tasks are really bad.

2. **Highly implausible evaluation results suggesting implementation errors.**
   The paper claims that even GPT-4o and Gemini-2.5-Pro can only solve linear systems of equations with two or three variables (line 1102-1108).  I cannot imagine anyone believing in this.  OpenAI and Anthropic are talking about getting IMO gold medals yet their models cannot solve linear equations.  Seriously?

   I asked GPT-4o to solve something like the following
   $$
   \\begin{cases}
   x + y + z + w + t = 15 \\\\
   2x - y + 3z + w - t = 10 \\\\
   x + 2y - z + 4w + t = 12 \\\\
   3x - y + 2z - w + 2t = 14 \\\\
   x - y + z + w + 3t = 16
   \\end{cases}
   $$
   and it does not seem to have a problem.

   Such extreme underperformance suggests that either (a) task formulation or parsing is flawed, or (b) the evaluation code is incorrect.  Because no code or dataset is available, these results cannot be reproduced or inspected or trusted.

   > **Actionable fix:** Release several failed examples and their model outputs, or share the generator code so that reviewers can verify whether the equations are malformed.  Please kindly provide 5 examples of three-variable linear equations that both GPT-4o and Gemini-2.5-pro fail to solve.

3. **Confusion between complexity and complication.**
   The authors equate *increasing task difficulty* with *adding more elements or loops* (e.g., deeper trees, larger graphs, more for-loops in code).  This confuses “complication” with “conceptual complexity.”  True **complexity** involves deeper abstraction or reasoning (e.g., algorithm synthesis, multi-step planning), not longer or more tedious or more **complicated** arithmetic.
   This same critique applies to prior work such as Shojaee et al. (2025) *“The Illusion of Thinking,”* later refuted by Opus & Lawsen (2025) for exactly this mistake.
   In that heavily criticised paper, Shojaee et al asked LMs to solve larger and larger Tower of Hanoi tasks until the LMs gave up writing down the detailed full solutions.  The point is that by just increasing the size, the Tower of Hanoi did not become more complex, deeper, harder, or more intellectually demanding; it just became more complicaited, tedious, and requiring patience to solve properly.   As shown in the refutation, the LMs can easily write down the correct recursive formulas.  “The generated solutions correctly implement the recursive algorithm, demonstrating intact reasoning capabilities when freed from exhaustive enumeration requirements.”

   Back to MACEval, the same error is repeated.  The tasks involved in paper are built in a similarly bad taste.  Instead of proposing genuinely more complex and more challenging tasks, the authors' operationalisation of controllable difficulty level is just making the task more complicated.  To wit,

- Instead of checking if the LM is able to propose and implement algorithms for binary tree traversal, MACEval just makes the tree more complicated.
- Instead of checking if the LM is able to propose and implement algorithms for shortest path search, MACEval just makes the graph more complicated.
- Instead of checking if the LM is able to generate useful and meaningful code for longer horizon realistic tasks, MACEval just adds arbitrary and pointless requirements like incorporating yet another for loop or while loop or if conditional statement.  A code generation task does not become meaningfully more complex if one introduces a dozen more for loops and another dozen more if conditional statements.  That is just jumping throw some arbitrary and useless hoops.

   In short, the “harder” tasks in MACEval are simply more verbose, not more cognitively demanding.

   > **Actionable fix:** Incorporate explicit *conceptual challenge dimensions* (e.g., compositional reasoning, ambiguity, multi-objective constraints) rather than numeric scaling of task size.  Otherwise, the tasks are not genuinely more difficult and do not represent meaningful future directions where we want to scale our future LMs.

4. **Low construct validity of the proposed tasks.**
   Drawing on classical measurement theory (Cronbach & Meehl, 1955), a benchmark must have clear construct validity—it should measure the intended capability.  In MACEval, task difficulty in code generation, for example, is operationalized as the number of `for`, `while`, or `if` statements in code, which does not reflect coding competence.  Experienced programmers know that debugging logical flow, managing dependencies, and ensuring semantic correctness—not loop count—define challenge.  Thus, the authors are misaligned with what the community cares about and want to evaluate.  There are so many meaningful and economically valuable aspects of coding that needs to be evaluated. One really should not pay attention to adding more `while` loops to coding task requirements.

   > **Actionable fix:** Re-define difficulty metrics based on semantic or reasoning depth (e.g., algorithmic novelty, multi-file context comprehension) rather than syntactic count.

5. **Presentation and clarity issues.**
   The command of English is overall good, but the math appears to be merely decorative and does not serve much narrative purposes.  After being defined it is not really used elsewhere and does not help with understanding.   This is just unnecessary cognitive load for a paper which suffers from clarity of expression.

   > **Actionable fix:** Simplify the math or actually use it.


**In summary**,
my question is this:
   > Why hasn't anyone published a benchmark containing lengthier and lengthier long-division and integer multiplication tasks?  (I am sure models all struggle with a task like calculating 182734018743091374 times 182347190387409138275, or any two huge numbers that an LM randomly comes up with.)

It is not because it is difficult to come up with such an idea, but because the idea is stupid.  The authors really need to explain why adding more `for` loops to a coding task requirement that they are doing is fundamentally different from my **dynamically-generated**, **extremely challenging**, **saturation-resistant**, **human-label-free**, **multi-agent**, long-division-and-multiplication benchmark generation pipeline.

**Questions:**

1. Could the authors release a subset of generated tasks (e.g., 20 examples across domains) to allow reviewers to inspect quality?
2. How are “difficulty levels” operationalized and verified, beside just "harder than before"?
3. Can the authors clarify the apparent inability of GPT-4o and Gemini-2.5-Pro to solve basic linear systems? This is really hard to believe.
4. How does MACEval ensure that generated tasks maintain *construct validity* rather than devolving into contrived puzzles?
5. Have human evaluators assessed whether tasks are meaningful or solvable? If not, do the authors plan to include such validation in a future version?

---

### Official Review · Reviewer_BfFa · 2025-11-01

**Soundness:** 2
**Presentation:** 3
**Contribution:** 2
**Rating:** 2
**Confidence:** 4

**Summary:**

This paper introduces a method of using LLMs to automatically generate evals and continuously change their difficulty.

**Strengths:**

Automated eval methods are promising and an important avenue of work. This paper proposes a workflow to achieve automated eval results for LLMs.

**Weaknesses:**

Missing (very) related work:
https://arxiv.org/pdf/2312.14856
https://arxiv.org/abs/2310.17567
https://arxiv.org/pdf/2502.06453
https://alignment.anthropic.com/2025/petri/

- It would be good to reflect on the quantity vs quality argument.
- This is presented as an automated eval method but humans have to manually come up with the tasks (only 9 introduced in the paper). E.g. construct an eval where you can iteratively add more noise to an image. While this is fine (the above linked papers do similar things) this paper goes to far in suggesting the full pipeline is automated.
- the 9 tasks are very simple, akin more to standard augmentation methods and not features which test frontier AI intelligence.
- Very little work was done to investigate how accurate the autograders were. Does a high score even correlate with intended behaviours/abilities? They have an ablation study measuring agreement between the autograders and humans but this was only on hallucination detection. The autograders have been tasked with doing considerably more complex tasks than this.
- over all this seems to be a very token hungry evaluation which yields hard to interpret results (what does it mean to be good at this eval?).

**Questions:**

see weaknesses

---

### Official Review · Reviewer_8Ctc · 2025-11-01

**Soundness:** 2
**Presentation:** 2
**Contribution:** 2
**Rating:** 2
**Confidence:** 4

**Summary:**

I actually read the paper carefully several times. I think the paper has several major flaws:

1. the authors need to significantly improve their presentation. As an AI researcher has done both benchmarking and agent paper myself, especially end-to-end agent self-evaluation frameworks, I almost could not understand the paper. The figures are not informative, the results are not enough, and the overall contribution is not clear in the paper.

2. The contribution of the paper is not clear. The authors propose a self-evaluation framework with agents, but could not clearly define the research gap, and the differences to exisiting works. Is the authors main contribution on the agent design side, or on the evaluation side, or both? How topology is helping the framework, what are the key diffeneces of the paper's benchmark, to, say, existing famous benchmark evaluation results, like in coding and math?

Overall, I feel the paper reads more like a tech report, not a research paper, especially at the ICLR level.

**Strengths:**

Good motivation

**Weaknesses:**

See summary

**Questions:**

Would suggest improve the contribution and the framework figures, like figure 1 and 2, too complicated and not clear at high level, not able to have take aways, solve all issues in summary

---

### Note · Authors · 2026-01-02

I have read and agree with the venue's withdrawal policy on behalf of myself and my co-authors.